# Evoked transients of pH-sensitive fluorescent false neurotransmitter reveal dopamine hot spots in the globus pallidus

Jozsef Meszaros[1,2], Timothy Cheung[3], Maya M Erler[4], Un Jung Kang[3], Dalibor Sames[5], Christoph Kellendonk[6,7,8]*, David Sulzer[3,6,7,8]*

[1]Laboratory for Functional Optical Imaging, Department of Biomedical Engineering, Mortimer B. Zuckerman Mind Brain Behavior Institute, Columbia University, New York, United States; [2]Graduate Program in Neurobiology and Behavior, Columbia University, New York, United States; [3]Department of Neurology, Columbia University, New York, United States; [4]Graduate Program in Pharmacology, College of Physicians and Surgeons, Columbia University, New York, United States; [5]Department of Chemistry and NeuroTechnology Center, Columbia University, New York, United States; [6]Division of Molecular Therapeutics, New York State Psychiatric Institute, New York, United States; [7]Department of Psychiatry, Columbia University, New York, United States; [8]Department of Pharmacology, Columbia University, New York, United States

**Abstract** Dopamine neurotransmission is suspected to play important physiological roles in multiple sparsely innervated brain nuclei, but there has not been a means to measure synaptic dopamine release in such regions. The globus pallidus externa (GPe) is a major locus in the basal ganglia that displays a sparse innervation of *en passant* dopamine axonal fibers. Due to the low levels of innervation that preclude electrochemical analysis, it is unknown if these axons engage in neurotransmission. To address this, we introduce an optical approach using a pH-sensitive fluorescent false neurotransmitter, FFN102, that exhibits increased fluorescence upon exocytosis from the acidic synaptic vesicle to the neutral extracellular milieu. In marked contrast to the striatum, FFN102 transients in the mouse GPe were spatially heterogeneous and smaller than in striatum with the exception of sparse hot spots. GPe transients were also significantly enhanced by high frequency stimulation. Our results support hot spots of dopamine release from substantia nigra axons.

DOI: https://doi.org/10.7554/eLife.42383.001

*For correspondence:
ck491@cumc.columbia.edu (CK);
ds43@cumc.columbia.edu (DS)

**Competing interests:** The authors declare that no competing interests exist.

## Introduction

Dopamine neurotransmission plays important roles in motor behavior, memory consolidation, reward prediction error and many other functions (*Matthews et al., 2016*; *Takeuchi et al., 2016*; *Sharpe et al., 2017*; *da Silva et al., 2018*). While the dopamine release from the dorsal and ventral striatal projections of ventral midbrain neurons is by far the most studied, there is also dopamine release from projections of neurons with cell bodies in the hypothalamus, dorsal raphe (*Matthews et al., 2016*) and the locus coeruleus (*Kempadoo et al., 2016*; *Sharpe et al., 2017*), and responses to dopamine in many regions, including the hippocampus (*Rosen et al., 2015*; *Kempadoo et al., 2016*), paraventricular thalamus (*Clark et al., 2017*; *Beas et al., 2018*), bed nucleus of the stria terminalis and the amygdala (*Matthews et al., 2016*). A range of approaches provide evidence for spatially-restricted, temporally-precise dopamine signaling (*Yagishita et al.,*

2014; *Howe and Dombeck, 2016*; *Bamford et al., 2018*). New optical approaches using fluorescent false neurotransmitters (FFNs) indicate that many dopaminergic axonal varicosities are functionally silent (*Pereira et al., 2016*; *Liu et al., 2018*), which may depend on the localization of molecular scaffolds (*Liu et al., 2018*), or presynaptic modulation by other neurons (*Threlfell et al., 2012*).

The GPe plays a significant role in shaping behavioral programs and initiating movement (*Mallet et al., 2016*; *Mastro et al., 2017*). Within the external globus pallidus (GPe), sparse dopamine axons were visualized in classic work using the glyoxylic acid method, which renders dopamine fluorescent in fixed tissue (*Lindvall and Björklund, 1979*). Branches of individual substantia nigra compacta (SNc) axons were visualized using a single cell viral infection protocol, demonstrating a relative paucity of axonal collaterals in the GPe (*Matsuda et al., 2009*).

It has not been known if sparse dopaminergic axons in GPe engage in dopamine neurotransmission. The level of dopamine release in the GPe is too sparse for analysis by carbon fiber amperometry and cyclic voltammetry, the principal methods for analyzing dopamine release in the heavily innervated dorsal and ventral striatum (nucleus accumbens). While very low levels of dopamine near detection threshold have been reported in the GPe in vivo by accruing a microdialysis sample over tens of minutes (*Hauber and Fuchs, 2000*; *Fuchs and Hauber, 2004*; *Hegeman et al., 2016*), there is no means to ascertain if this is due to GPe intrinsic neurotransmission or overflow from the highly innervated neighboring striatum.

We have an ongoing effort to develop FFNs to visualize neurotransmitter uptake and release in living brain tissue and in vivo. In this study, we adapt a pH-sensitive FFN, FFN102, which is a substrate for the dopamine transporter, DAT, and the vesicular monoamine transporter, VMAT2 (*Rodriguez et al., 2013*) as the first use of a 'flashing FFN'. The signal from FFN102 is well-suited for studying synaptic release in sparsely innervated regions as it is brighter in the neutral extracellular milieu than the acidic milieu of the synaptic vesicle lumen (*Lee et al., 2010*). We find that upon electrical stimulation, FFN102 produces a calcium-dependent flash of fluorescence that requires functional dopamine fibers emanating from the SNc. The release properties differ from the striatum independently of DAT activity. These results indicate that pH-sensitive FFNs provide a means to study dopamine release within brain areas of sparse dopaminergic innervation.

## Results

### FFN102 release differs between the GPe and striatum

Fluorescent false neurotransmitters have previously been used to directly measure dopamine synaptic vesicle fusion and exocytosis from individual *en passant* release sites on the axon, termed puncta (*Gubernator et al., 2009*; *Pereira et al., 2016*). These FFN methods require tracking fluorescence within micron-sized regions in a field of view. Such methods use z-stacks to track puncta, which lowers temporal resolution. In these experiments, individual electrical pulses produce relatively small changes in fluorescence, and experimenters had to apply hundreds of pulses in order to generate a measurable signal.

As an alternate approach for sparsely innervated regions, we have adapted FFN102, a pH-sensitive fluorescent false neurotransmitter that is a substrate for the dopamine uptake transporter and the vesicular monoamine transporter and exhibits higher fluorescent emissia at extracellular neutral pH than the acidic synaptic vesicle pH (*Lee et al., 2010*; *Rodriguez et al., 2013*). As dopamine innervation in the GPe is very sparse, there are few FFN-labeled structures in a given field of view. We thus chose to average all pixels within each frame to provide a whole-field fluorescence measurement. In contrast to the endocytotic synaptic vesicle dye, FM1-43, FFN102 enters synaptic vesicles as a transporter substrate without electrical stimulation (*Rodriguez et al., 2013*). We thus used a 30 min incubation period without electrical stimulation to load cells with the probe (*Figure 1A*). To minimally disturb the synapse and allow for examining plasticity and modulation, we chose a stimulation paradigm that employed a brief stimulus period with an applied stimulus current of 200 μA. A bipolar electrode was placed on the slice and oriented so that the two poles contacted the dorsal and ventral aspects of the GPe. Fields of view imaged were 50 to 100 μm from one of the two electrode poles. We selected a stimulus frequency of 10 Hz, as it provided consistent responses and is within the range of dopamine neuron burst firing in vivo (*Paladini and Roeper, 2014*).

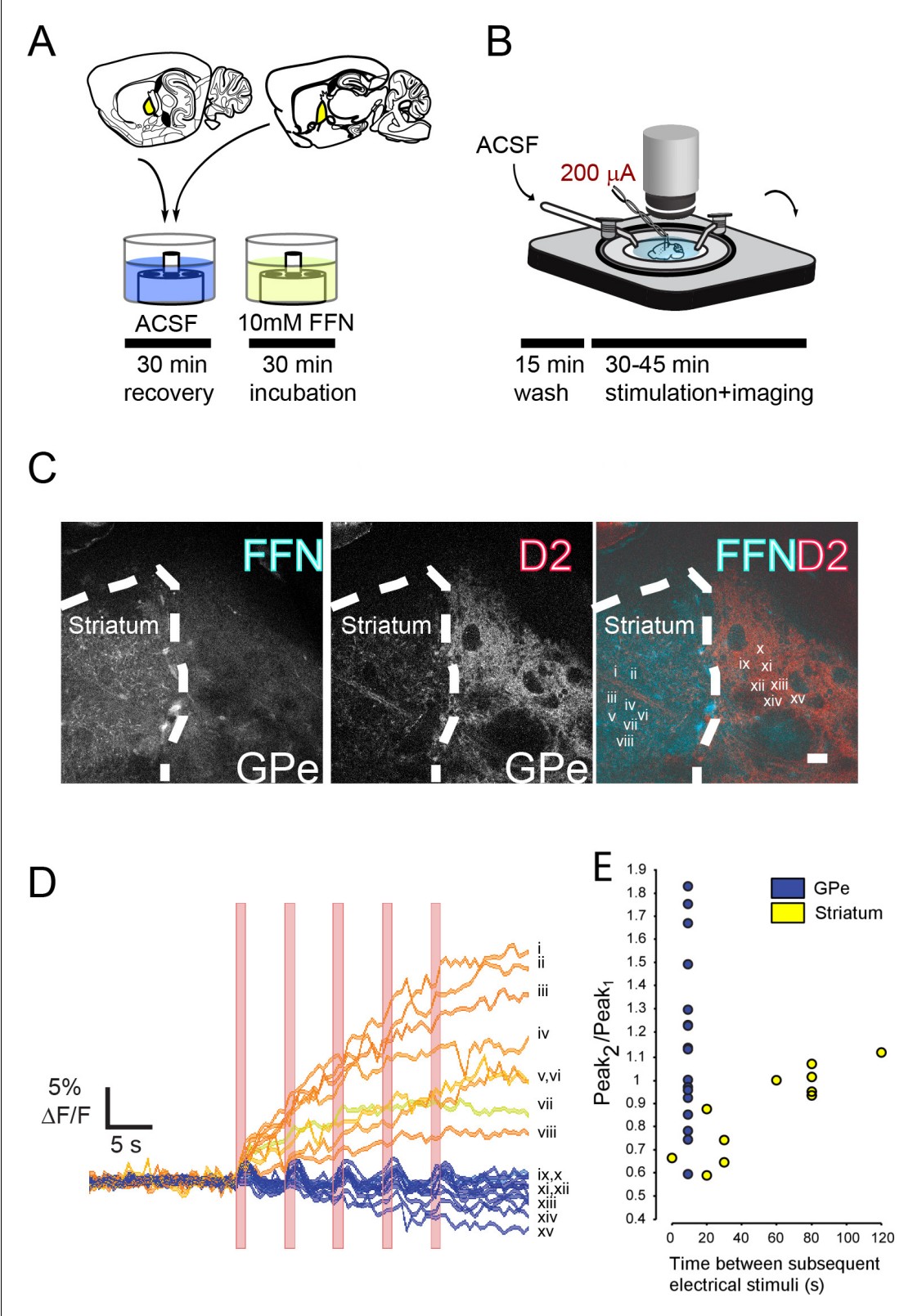

**Figure 1.** Electrical stimulation of GPe evokes FFN102 transients. (**A, B**) Preparation of GPe brain slice. (**C**) In BAC-D2 GFP mice, the striatum and GPe are distinguishable, as the GPe receives a thick plexus of D2-positive terminals, while the striatum is rich in FFN labeled processes. Scale bar = 50 μM. (**D**) In response to 1 s long electrical stimulation at 10 Hz frequency, FFN fluorescence corresponding to regions in panel C) show a 'flashing' pattern of transients in the GPe (inset Roman numerals for traces ix-xv) and a prolonged and sustained increase in fluorescence in the striatum (inset Roman

*Figure 1 continued on next page*

*Figure 1 continued*

numerals for traces i-viii). Bars indicate the period of each stimulus (1 s). (E) Signal amplitude of stimuli at one interval in the GPe (from data in panel D), and a range of intervals in the striatum (from a different experiment): note the high variability of signal in GPe.

DOI: https://doi.org/10.7554/eLife.42383.002

To determine if the FFN release was localized within the GPe, and not due to diffusion from the striatum, we compared electrically stimulated changes in fluorescence within the striatum and GPe. To clearly delineate the boundary between the two areas, we used Drd2-BAC-GFP mice (*Figure 1C*): many D2 medium spiny neuron terminals converge within the GPe, creating a dense field of GFP fluorescence. While FFN102 labeled presynaptic elements profusely innervated the striatum, there were few obvious puncta within the GPe (*Figure 1C*, leftmost panel). When the slice was stimulated, the rapid alkalization of exocytosed FFN102 increased the whole-field fluorescence. The resulting fluorescence intensity profiles for these regions of interest are shown as red/pink lines and numbered i-viii for imaging within the striatum, and as blue lines numbered ix-xv in the GPe.

We observed different kinetics of FFN transients in the two regions. First, in the striatum, the fluorescence intensity remained elevated over time, while in the GPe the signal was much smaller and much shorter in duration. Second, in the striatum, pairs of stimuli needed to be separated by at least 60 s for the second transient to recover a signal comparable to the first, similar to the rate of recovery for dopamine release measured with cyclic voltammetry (*Schmitz et al., 2002*); in contrast, in the GPe, ten second intervals were often sufficient to recover the full response.

We performed control experiments to confirm if we were measuring release from dopaminergic terminals. First, as electrical stimulation can produce fluorescence changes through NADH metabolism in some in vitro preparations (*Kasischke et al., 2004*), we measured fluorescence changes in response to electrical stimulation in slices pre-incubated with or without FFN. To compare FFN transients, we averaged responses to multiple trains of stimuli in a field of view to obtain a slice-average FFN transient. Slices that had not been incubated with FFN showed no transient, measured as the area under the curve (AUC) (*Figure 2A,B*). Thus, FFN release, and not another process, was responsible for the transient events.

To examine if FFN102 is loaded in GPe axons as a substrate for the dopamine transporter (DAT), we co-incubated slices with both FFN102 and the DAT inhibitor, nomifensine, for 30 min before the stimuli. The significant decrease of evoked fluorescent transients in slices co-incubated with nomifensine is consistent with an inhibition of uptake and loading of FFN102 into DA axons (*Figure 2C,D*).

## FFN102 transients reflect synaptic dopamine release

To assess whether the release of FFN102 within the GPe was of synaptic origin, we analyzed the effect of extracellular $Ca^{2+}$ on FFN transient size. For these experiments, the levels of calcium reaching a slice were randomly alternated between 0.5, 2, and 4 mM $Ca^{2+}$. An individual slice was perfused with a given concentration, allowing ten minutes for the calcium concentration to adjust within the slice. There was main effect of calcium levels on the AUC %ΔF/F (repeated measures ANOVA, $p < 0.001$) (*Figure 3*). Additionally, there was a significant difference between AUCs in the striatum and GPe ($p < 0.001$), as expected from our other experiments. We also found a significant interaction between calcium and region ($p < 0.05$). Consistent with release by synaptic vesicle fusion from dopamine axons in the striatum, 0.5 mM extracellular calcium produced significantly smaller average FFN102 striatal transients than 2 mM or 4 mM (*Figure 3A,B*). Similarly, within the GPe, calcium enhanced FFN102 transients (*Figure 3C,D*), with the response to increased calcium apparently saturating above 2 mM.

To confirm whether dopaminergic neurons were the source of the FFN transients, we used the toxin 6-hydroxydopamine (6-OHDA lesion) to unilaterally lesion the dopamine projections passing through the medial forebrain bundle. We used tyrosine hydroxylase (TH) immunolabel to confirm the lesions. We subsequently prepared slices from both hemispheres and recorded FFN transients within each slice (*Figure 4A,B*). The global averages of FFN transients for the lesioned and unlesioned sides were compared. Consistent with the hypothesis that FFN is released from DA neurons, 6-OHDA lesion significantly decreased the FFN transients (*Figure 4B,C*). After imaging FFN transients, we post-fixed and confirmed the lesion in the slices used for imaging. We noted that FFN transients

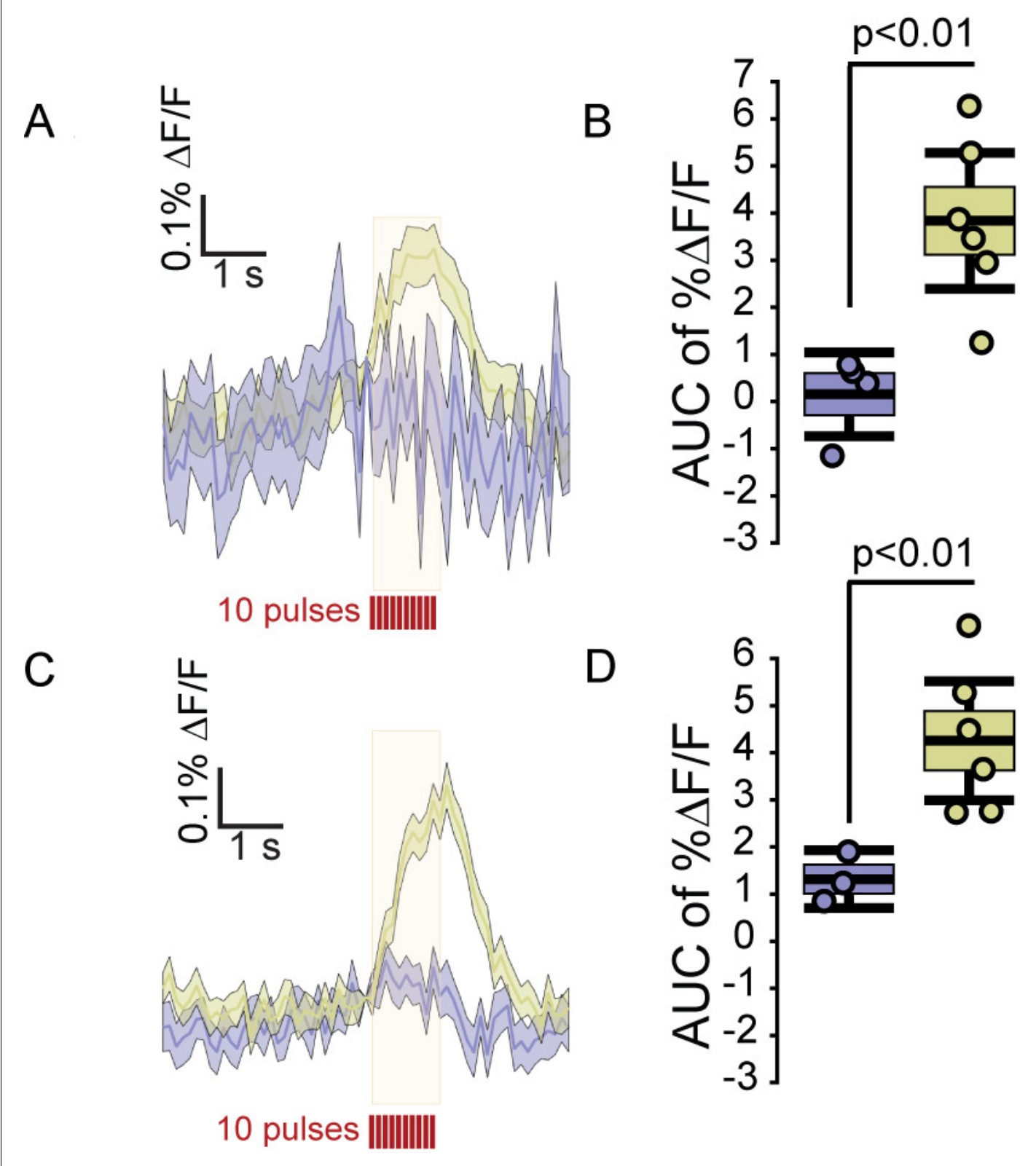

**Figure 2.** FFN102 and DAT dependence of fluorescence transients. (**A**) Averaged traces of electrically evoked fluorescence from slices incubated with FFN, gold, or ACSF alone, blue. Traces reflect the average %ΔF/F, shading reflects ±SEM (n = 4 slices for ACSF, n = 6 slices for 10 μM FFN). (**B**) Box-and-whisker plot showing the AUC %ΔF/F evoked in the FFN and ACSF conditions. The edges of the boxes are 1 SEM from the mean, and the whiskers indicate ±2 SEM. Each point represents the average of images from one slice. ACSF slices were significantly different from FFN slices (two-tailed

*Figure 2 continued on next page*

Figure 2 continued

unpaired t-test, p < 0.01). For ACSF slices, the AUC %ΔF/F was 0.1 (CI$_{95}$=[−0.78,0.98]) compared to 3.8 for FFN slices (CI$_{95}$=[2.31,5.29]). (C) Transients from slices that were either incubated with FFN, gold, or FFN with 10 μM nomifensine, blue (n = 3 slices for FFN with nomifensine and n = 6 slices for FFN alone). (D) FFN102 transients evoked by electrical stimulation were strongly decreased by nomifensine (two-tailed unpaired t-test, p < 0.01; FFN, mean AUC %ΔF/F = 1.3 (CI$_{95}$ = [0.61,1.99]) FFN with nomifensine, mean AUC %ΔF/F = 4.12 (CI$_{95}$ = [2.97,5.27]).

DOI: https://doi.org/10.7554/eLife.42383.003

in the non-lesioned side were decreased relative to control slices. Therefore, 6-OHDA depletion in one hemisphere may lead to a reduction of the GPe FFN transient on the contralateral side, consistent with reports that contralateral dopamine projections extend to the GPe (*Pritzel et al., 1983*; *Douglas et al., 1987*).

To assess the anatomical source of the evoked transients, we turned to an aphakia mouse line, in which mutation of the pitx3 gene prevents the development of dopamine neurons selectively in the substantia nigra pars compacta (SNc) while sparing VTA dopamine neurons (*Nunes et al., 2003*; *van den Munckhof et al., 2003*). We confirmed this depletion in our mice using TH immunolabel (*Figure 4D*). To obtain within-animal comparisons for aphakia mice, we randomly interleaved unstimulated 'sham' imaging epochs during which we collected images without electrical stimulation. We found that wild type mice had significantly larger FFN transients than aphakia mice (*Figure 4E*). The aphakia slices showed a small but significant difference between stimulated and unstimulated experiments, possibly arising from spared VTA neurons (*Figure 4F*). Thus, SNc dopamine neurons are the primary contributor to FFN102 release in the GPe.

## FFN102 release reveals differences in dopamine release between striatum and GPe

To examine the frequency dependence of FFN102 release in striatal and GPe dopamine synapses, we evoked FFN transients with 10 Hz and 50 Hz trains. We observed a main effect of frequency on FFN transients which was mostly due to the effects on GPe (two-factor ANOVA, p<0.001). Within the striatum, we did not observe significant differences in FFN transient size at 10 Hz and 50 Hz stimuli (*Figure 5A*). In contrast, within the GPe, the area under the curve for the transient evoked by 50 Hz was significantly higher than for 10 Hz (*Figure 5B,C*).

We then addressed whether the differences in kinetics and size of FFN transients between the striatum and the GPe might result from differential reuptake, which might broaden the signal. For these experiments, we performed within-slice paired measurements of FFN transients in ACSF alone and after 10 min of perfusion with 10 μM nomifensine-containing ACSF. Nomifensine slightly decreased the FFN transient size in the striatum (*Figure 6A,B*), but not in the GPe (*Figure 6C,D*). Neither the decay time to its half-maximum value (*Figure 6E*) nor the shape of the decay were altered by nomifensine (*Figure 6F*), indicating a lack of a role for DAT in altering evoked FFN102 signals.

The greater release of FFN102 in GPe at higher frequencies (*Figure 5*) suggested that GPe dopamine axons might have the capacity to generate large FFN transients. Indeed, we occasionally observed transients in the GPe of comparable size to striatum (*Figure 7A*). Surprisingly, GPe regions with large FFN102 transients did not exhibit obvious FFN102 puncta within the field of view (*Figure 7B*). We then examined if the large GPe transients were due to the presence of any puncta. To do so, we measured AUC values for transients and their correlation with the Canny Edge sum, a measure of high contrast edges of an image, specifically those around fluorescent puncta and neuropil (*Figure 7C*). We found very little correlation between GPe transient magnitude and Canny Edge values (R = 0.051) and found a similarly low correlation between GPe transient magnitude and initial image fluorescence (R = −0.062). Thus, neither the number of puncta nor the initial image fluorescence appeared to be related to the variability in FFN transient size.

We next examined whether the regions of high release were spatially clustered. We found that GPe slices rarely contained more than one such 'hotspot' of FFN release (*Figure 7D,E*). To quantify these observations, we segregated fields of view from a slice, assuming each field contains a putative release area, into four equal clusters based on the AUC from that field of view. We measured

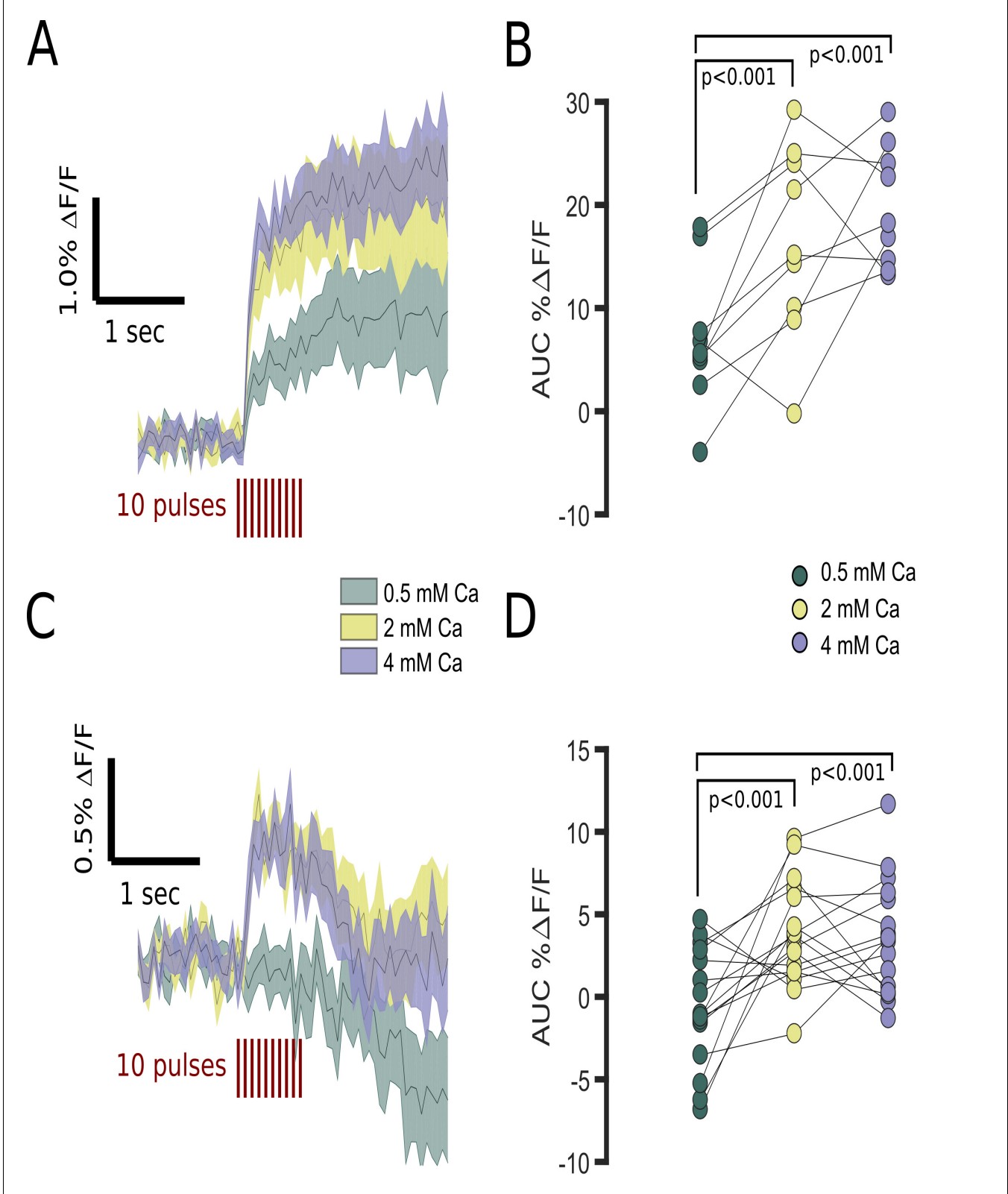

**Figure 3.** FFN102 transients are regulated by extracellular calcium. (**A**) Average processed transients of electrically evoked transients from striatal areas in slices perfused with 0.5, 2, and 4 mM $Ca^{2+}$ (n = 9 slices). (**B**) There was a main effect of calcium-level on AUC %ΔF/F (repeated measures ANOVA, p < 0.001). The AUC %ΔF/F was higher in 2 and 4 mM than 0.5 mM $Ca^{2+}$ (two-tailed paired t-test, p < 0.001). (**C**) Average processed transients of
*Figure 3 continued on next page*

*Figure 3 continued*

electrically evoked transients from GPe in slices perfused with 0.5, 2, and 4 mM Ca$^{2+}$ (n = 16 slices). (**D**) The AUC %ΔF/F was higher in 2 and 4 mM than 0.5 mM Ca$^{2+}$ (two-tailed paired t-test, p < 0.001). Additionally, there was a significant interaction between calcium levels and region (p<0.05).

DOI: https://doi.org/10.7554/eLife.42383.004

The following source data and source codes are available for figure 3:

**Source data 1.** Fluorescence time courses for FFN transients evoked under varying calcium levels.
DOI: https://doi.org/10.7554/eLife.42383.005
**Source code 1.** Analysis and figures for FFN102 transients under varying calcium levels.
DOI: https://doi.org/10.7554/eLife.42383.006

the distance between putative release areas (*Figure 7F*). If the hotspots were clustered, we would expect that the distribution of the top quarter of fields of view to have a small distance between pairs of release sites. We found that the most active areas were not closely spaced, and were as likely to be neighbors with a low releasing area as a high releasing area.

To compare the response to each electrical pulse between the GPe and striatum, we analyzed the derivative of the FFN transients, binned to average the intensity for frames corresponding to each electrical pulse. The averaged derivative for all regions of striatum and GPe was a sharp, single peak (*Figure 7G*), with a larger striatal peak due to the larger transient. We then determined each field of view's derivative value before stimuli, after a single pulse, and after two pulses (*Figure 7H*). While we only resolve signals at 100 msec intervals, it is apparent that for both the striatum and the GPe, the largest contribution to the total transient occurred at the first electrical pulse and is consistent with estimates of the rising phase using cyclic voltammetry, which is ~180 msec in the striatum (*Schmitz et al., 2001*).

## Discussion

A variety of FFNs now provide means to observe release during synaptic vesicle fusion from specific sites on axons, including striatal dopamine axons in vitro (*Gubernator et al., 2009*; *Rodriguez et al., 2013*; *Pereira et al., 2016*) and norepinephrine axons in vitro and in vivo (*Dunn et al., 2018*). Here, we introduce a new use for pH sensitive FFNs, which is to resolve release from dopamine axons in a region of very sparse innervation by 'FFN flashes', that is short duration calcium-dependent release transients during exocytosis as the fluorescence is unquenched upon exocytosis from the acidic vesicle lumen to the neutral extracellular space. FFN102, a DAT and VMAT2 substrate with a pKa of 6.2 (*Lee et al., 2010*), is well suited for this application, as its signal increases nearly 400% between the acidic synaptic vesicle (~pH 5.6) and the extracellular milieu at pH 7.4 (*Rodriguez et al., 2013*).

As might be expected from the sparse innervation by dopamine axons in the GPe, the same electrical stimulus generally elicited far smaller FFN transients in the GPe than the striatum, but occasional GPe 'hotspots', about one field of view per GPe slice, were found in which the level of evoked release was similar. The GPe hotspots had no clear lateral-to-medial pattern and were spatially segregated. Such hotspots of release have been described within the striatum as regions that include dopamine diffusing from active, but distant, sites of release (*May and Wightman, 1989*; *Rodriguez et al., 2006*). Alternatively, dopamine hotspots may be present where other modulatory cells are present, for example cholinergic cells, of which a few have been observed in the GPe (*Gielow and Zaborszky, 2017*). The identification of hotspots in the GPe is consistent with recent work showing that many striatal dopamine varicosities with clusters of synaptic vesicles are silent (*Pereira et al., 2016*), and this could be due to the absence or presence of local presynaptic scaffolding proteins (*Liu et al., 2018*). The GPe hotspots might represent areas where multiple collateral dopamine axons converge or areas with enriched presynaptic proteins. However, these active loci apparently do not contain large clusters of VMAT2-expressing vesicles observed after FFN102 loading. We also note that electrical stimulus of the striatum also stimulates striatal cholinergic interneurons, which induce additional release (*Melchior et al., 2015*), and that is expected to play far less of a role in the GPe, which is not known to have intrinsic cholinergic neurons. Local interactions with cholinergic interneurons may contribute to differences in the FFN signals between striatum and GPe,

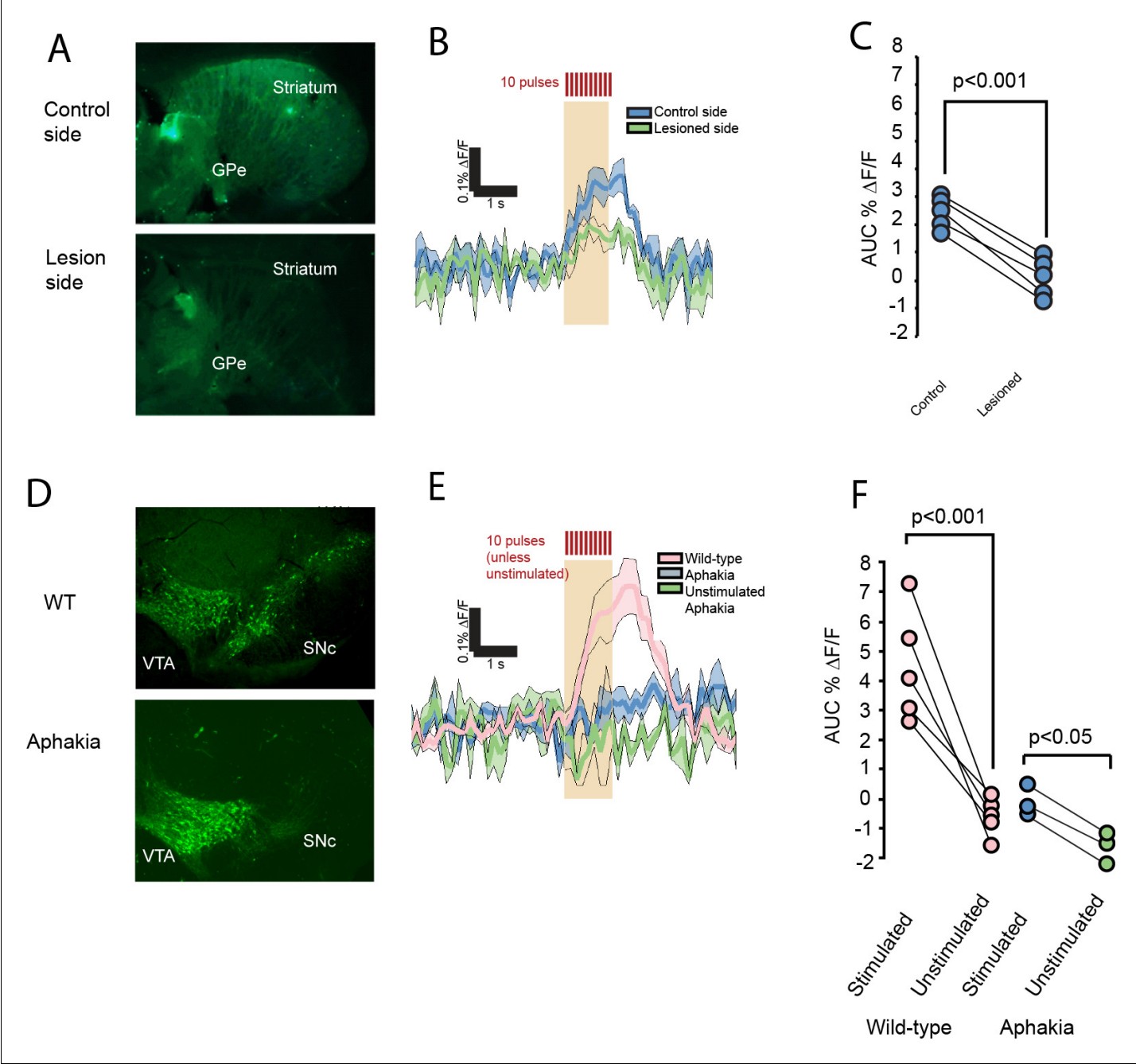

**Figure 4.** FFN102 transients are ablated in dopamine depleted mice. (**A**) Representative images of slices in 6-OHDA experiments, post-fixed after imaging and immunolabeled for tyrosine hydroxylase. (**B**) Evoked transients from control side, blue, or lesioned side, green (n = 5 slices from five mice for each condition; shading represents SEM). (**C**) The lesioned hemispheres show decreased transients (average difference of 2.46, $CI_{95}$ = [2.10, 2.82], two-tailed paired t-test p < 0.001). (**D**) Representative immunolabel for tyrosine hydroxylase in wild-type and aphakia mice shows a decreased label in the SNc, but not VTA. (**E**) Evoked transients wild-type mice of the same background, pink, aphakia mice, blue, and unstimulated aphakia mice, green (shading represents SEM). (**F**) Comparison of stimulation-evoked fluorescence changes in aphakia and wild-type mice (average difference in AUC %ΔF/ F = 4.54 ($CI_{95}$ = [3.05, 6.03], n = 5 slices from five mice for WT experiments, p < 0.001); between stimulated and unstimulated slices from aphakia mice, the average difference in the AUC %ΔF/F = 1.46 ($CI_{95}$ of difference = [1.00, 1.91]; n = 3 slices from three mice for aphakia experiments; p < 0.05).
DOI: https://doi.org/10.7554/eLife.42383.007

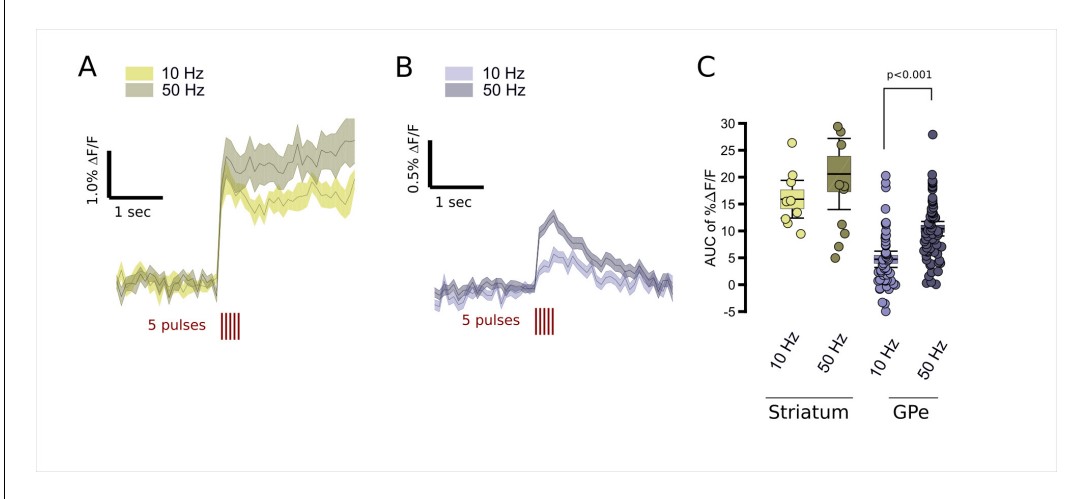

**Figure 5.** Modulation of FFN102 transients by stimulus frequency. (**A**) FFN transients evoked in the striatum by five pulses (red lines) at 10 Hz, gold traces, or 50 Hz, dark gray traces (N = 9 slices for 10 Hz stimuli, N = 12 slices for 50 Hz stimuli; shading represents SEM). (**B**) FFN transients evoked in the GPe by five pulses at either 10 Hz, bright blue traces, or 50 Hz, dark blue traces (N = 52 slices for 10 Hz stimulation, N = 68 slices for 50 Hz stimulation). (**C**) The AUC %ΔF/F for the period from 0 to 300 ms from stimulus onset. Striatal responses were not significantly different at higher frequencies (CI$_{95}$ of %ΔF/F at 10 Hz = [12.6,19.1] versus at 50 Hz = [14.7,26.5]), whereas GPe showed significantly higher AUC %ΔF/F at 50 Hz than 10 Hz (CI$_{95}$ of %ΔF/F at 10 Hz = [3.46, 5.99] versus at 50 Hz = [9.28, 11.5]), significance assessed using two-tailed unpaired t-test, p < 0.001.

DOI: https://doi.org/10.7554/eLife.42383.008

The following source data and source codes are available for figure 5:

**Source data 1.** Fluorescence time courses for FFN transients evoked by 10 Hz and 50 Hz stimulation.
DOI: https://doi.org/10.7554/eLife.42383.009
**Source code 1.** Analysis and figures for FFN102 transients evoked by 10 Hz and 50 Hz stimulation.
DOI: https://doi.org/10.7554/eLife.42383.010

although these effects are minimized with the train stimuli used in this study (*Rice and Cragg, 2004*; *Zhang and Sulzer, 2004*; *Cachope et al., 2012*; *Melchior et al., 2015*).

While striatal FFN transients were easily measured in the striatum in response to a single stimulus, the much smaller FFN transients in the GPe were most effectively detected with a train of stimuli. In marked contrast to the striatum, the FFN transient in the GPe decayed rapidly (*Figure 6E,F*) consistent with diffusion from a small amount of release sites, and appeared as a 'flash'. We used trains of 10 pulses to release sufficient FFN, which prolongs the signal. Most GPe regions showed release after the first pulse, but the amount of evoked release was strikingly variable, and far more responsive to increased stimulus frequency than release in the striatum. The amplitude of the GPe FFN transients was not correlated with the number of puncta in the images (*Figure 7C*). A recent ultrastructural study of dopamine axons in the striatum indicates that dopamine synaptic vesicles are present throughout the axon, with synaptic vesicle clusters at identifiable synapses (*Gaugler et al., 2012*). The lower amplitude evoked FFN transients in GPe may be due to exocytosis from comparatively small reservoirs of synaptic vesicles in thin portions of the axon that do not maintain large vesicle clusters. Indeed, the sparse observable puncta in GPe (i.e., apparent boutons with FFN loaded vesicles) are functionally silent (in the striatum, ~80% of puncta are inactive under electrical stimulation). In the current work we discovered that the FFN release occurs primarily at sites that do not contain clusters of vesicles observable by FFN imaging.

The decay of FFN102 transients to the apparent detection limit was much faster in the GPe than the striatum, including in GPe hotspots with high evoked transient amplitudes similar to the striatum. The GPe and other sparsely innervated regions have little or no dopamine uptake transporter (DAT) activity (*Miller et al., 1997*). Differences in reuptake, however, were not responsible for the difference in transient kinetics, as while DAT blockade during FFN102 incubation blocked transients by inhibiting loading (*Figure 2*), DAT blockade during the stimuli did not affect the transients in either striatum or GPe (*Figure 6*). The longer fluorescence decay in striatum is in part due to diffusion from

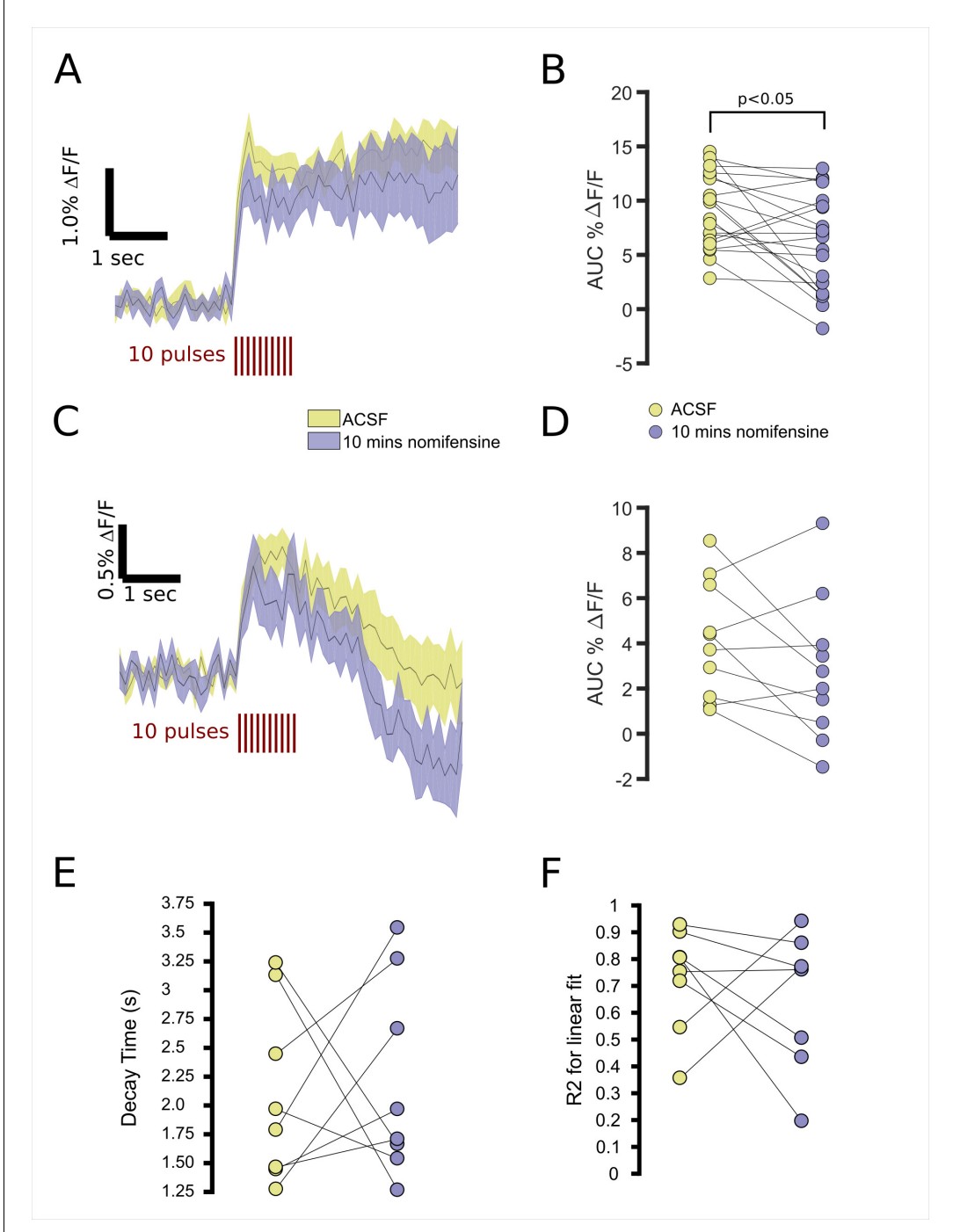

**Figure 6.** Differences in FFN102 transients in striatum and GPe are unrelated to reuptake. (**A**) FFN transients imaged in the striatum for slices perfused first with ACSF alone followed by ten minutes with nomifensine. (**B**) The average AUC during the stimulus was 8.56 for control slices with a CI95 = [6.92,10.20], and for nomifensine, the average AUC was 5.90 with a $CI_{95}$ = [3.94, 7.85] (paired t-test, p < 0.05). (**C**) FFN transients imaged in the GPe for slices similarly treated. (**D**) The average AUC during the stimulation time period was 4.16 with a $CI_{95}$ = [2.56, 5.75], and for nomifensine, the average AUC was 2.78 with a $CI_{95}$ = [0.81, 4.75] (paired t-test, p > 0.05). (**E**) Decay constants of log-transformed GPe traces show the time for the transient to decay to 10% of its initial value. Treated slices had an average decay time of 2.09 s, $CI_{95}$= [1.56, 2.62], and untreated slices had an average of 2.20 s, $CI_{95}$= [1.61, 2.79]. (**F**) Goodness-of-fit measurements for log-transformed traces.

DOI: https://doi.org/10.7554/eLife.42383.011

The following source data and source codes are available for figure 6:

**Source data 1.** Fluorescence time courses for FFN transients evoked under nomifensine block.

*Figure 6 continued*

DOI: https://doi.org/10.7554/eLife.42383.012

**Source code 1.** Analysis and figures for FFN102 transients during nomifensine block.

DOI: https://doi.org/10.7554/eLife.42383.013

out-of-plane striatal terminals, whereas in the GPe, areas of high release are few and spatially separated, and so less signal would diffuse from distal release sites to the regions of interest (*Sulzer and Pothos, 2000*).

While tracing studies show that SNc neurons pass through the GPe (*Matsuda et al., 2009*), there could be additional sources of GPe dopamine release. For example, recent work highlights dopaminergic dorsal raphe cell inputs to the amygdala (*Matthews et al., 2016*) and locus coeruleus inputs to the hippocampus (*Kempadoo et al., 2016*). By specifically depleting SNc DA axons using toxins, and examining the aphakia mouse line in which SNc DA neurons are absent and dopamine release in dorsal striatum is abolished (*Lieberman et al., 2018*), we confirmed that the axons that release FFN102 are dopaminergic fibers that emanate from the SNc (*Figure 4*).

The use of pH-sensitive 'flashing' FFNs as optical analogs to cyclic voltammetry and amperometry for the detection of catecholamine neurotransmission in regions of low innervation may prove to be a useful experimental tool. The current approach is able to resolve FFN transients in $30 \times 30$ μm fields of view, close to the size of neuronal cell bodies, and so FFN transients might be used to characterize dopamine release near specific cell types, such as the arkypallidal cells which project back into the striatum and the protopallidal cells that project to the SNr and GPi (*Gittis et al., 2014*; *Mastro et al., 2014*; *Hernández et al., 2015*). Dopamine is known to disinhibit GPe cells (*Cooper and Stanford, 2001*; *Shin et al., 2003*), and could modulate the balance between the arky- and protopallidal pathways. The recently introduced FFN270, which is a substrate for the norepinephrine transporter (*Dunn et al., 2018*), is also pH sensitive and could provide means to measure properties of the sparse and widely distributed network of norepinephrine axons in the nervous system.

# Materials and methods

## Key resources table

| Reagent type (species) or resource | Designation | Source or reference | Identifiers | Additional information |
|---|---|---|---|---|
| Strain, strain background (*Mus musculus*) | Male and female C57BL/6J mice | The Jackson Laboratory | RRID:IMSR_ JAX:000664 | |
| Strain, strain background (*Mus musculus*) | Male Drd2-BAC-GFP mice | MMRC | MGI:3843608 | |
| Strain, strain background (*Mus musculus*) | Male and female aphakia mice | Provided by Dr. Un Kang (doi:10.1111/gbb.12210; doi: 10.1073/pnas.1006511108) | | |
| Antibody | mouse monoclonal anti-tyrosine hydroxylase | Millipore | RRID:AB_390204 | (1:750) |

## Ethics statement

All animal protocols followed NIH guidelines and were approved by Columbia University's Institutional Animal Care and Use Committee (protocol AC-AAAR4420).

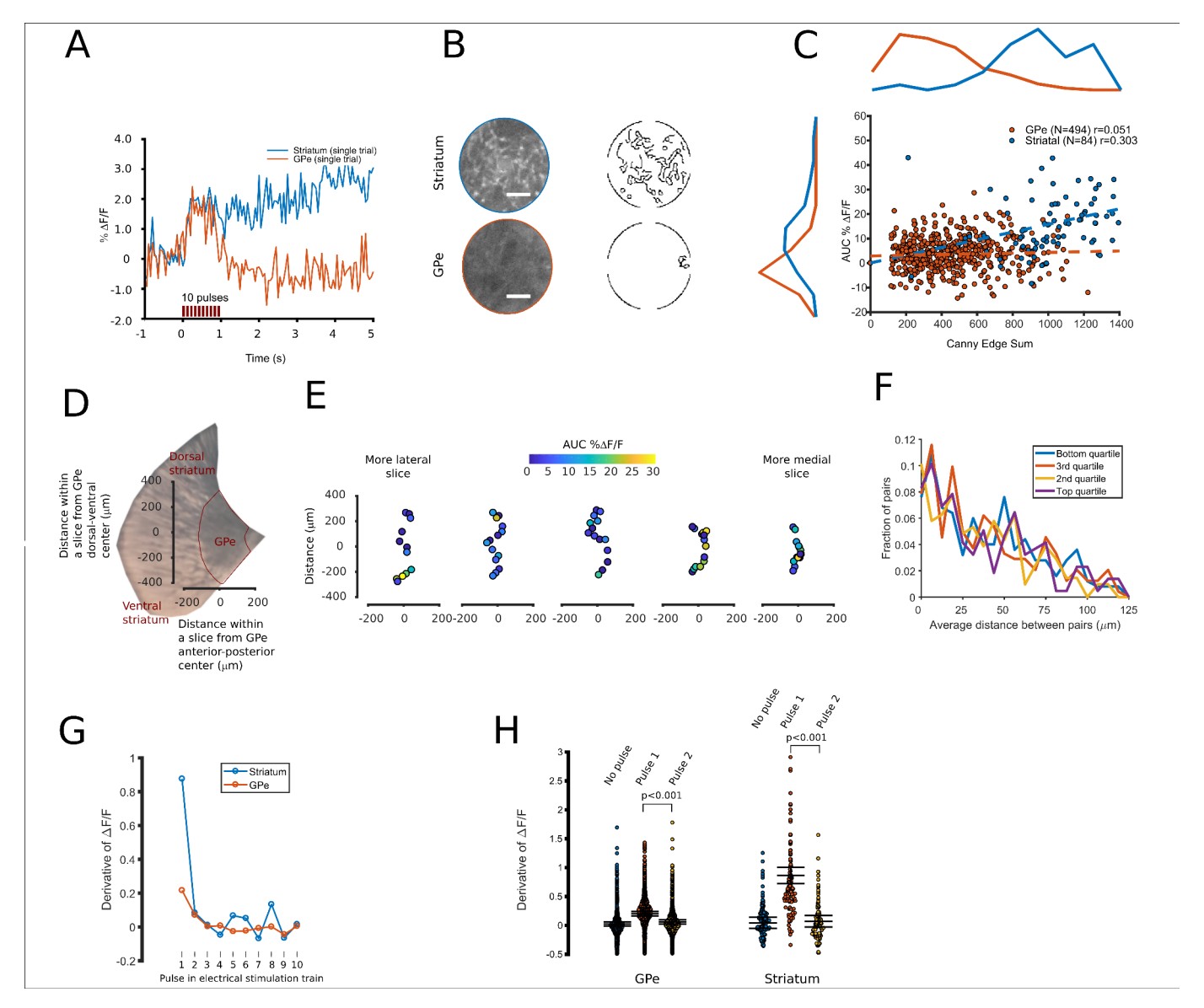

**Figure 7.** Spatial and temporal characterization of FFN102 transients. (**A**) A representative FFN transient from striatum, blue, and a 'hotspot' within the GPe, red. (**B**) Canny filtered masks were calculated from the average of each field of view's baseline images, returning a pixel value of '1' if the area has a high contrast and otherwise '0'. The fields of view that produced these transients are shown, both in their raw form and as a Canny edge filtered image. White scale bar = 10 μm. (**C**) For each field of view, the Canny edge sum (x axis) is displayed with the amplitude of its FFN transient (y axis). Correlation values are shown for GPe (n = 494 fields of view) and striatum (n = 84 fields of view). (**D**) An area of a representative brain slice containing the GPe. The two axes display distance in μm along the long axis of the GPe. (**E**) Points oriented to show imaging locations along the long axis of the GPe, with colors corresponding to their AUCs. Slices are shown from left to right, from more lateral to more medial slices. (**F**) Histograms showing the distance between pairs of all fields of view imaged within a slice. The distributions are split into four quartiles, where pairs in the top quartile both had the largest FFN transients compared to other pairs in the same slice. Similarly, pairs in the bottom quartile both had smaller FFN transients compared to the other 75% of pairs. (**G**) A plot of the derivative of the fluorescence intensity over time, averaged over the fields of view for each region (N = 84 for striatum, N = 494 for GPe). (**H**) Values of the derivative of the fluorescence intensity over time, for three intervals: 100 ms prior to stimulation, 100 ms after the first pulse, and 100 ms after the second pulse. Error bars represent the mean with CI95. All non-overlapping error bars were significant at p < 0.001. The mean value for the derivative at the first pulse was 0.86% for the striatum and 0.20% for the GPe.

DOI: https://doi.org/10.7554/eLife.42383.014

The following source data and source codes are available for figure 7:

**Source data 1.** Fluorescence time courses for spatial and temporal analysis.
DOI: https://doi.org/10.7554/eLife.42383.015

*Figure 7 continued*

**Source code 1.** Analysis and figures for FFN102 transients used for spatial and temporal analysis.
DOI: https://doi.org/10.7554/eLife.42383.016

## Mice

Experiments were performed on male and female C57BL/6J mice (9–24 weeks old) obtained from Jackson Laboratories (Bar Harbor, ME, USA), Drd2-BAC-GFP (S118Gsat/Mmnc) mice purchased from MMRRC, and *aphakia* mice. The *aphakia* allele is a loss-of function mutation in the Pitx3 gene arose spontaneously on the 129/Sv-S1 j strain at the Jackson Laboratory and was maintained in a C57BL background as previously described (*Ding et al., 2007*; *Ding et al., 2011*). All mice were housed under a 12 hr light/dark cycle in a temperature-controlled environment with food and water available ad libitum.

## 6-OHDA lesion

Unilateral 6-hydroxydopamine (6-OHDA) lesions were performed in male and female C57/BL6 mice of 12–15 weeks of age. Mice were anesthetized with ketamine/xylazine cocktail (80 mg/kg ketamine and 4 mg/kg xylazine, i.p.). Mice received unilateral lesion of the left medial forebrain bundle by intracranial infusion of 4.5 μg of 6-OHDA free base in 1.5 μl of 0.05% ascorbic in 0.9% saline at a rate of 300 s/μl into the following coordinates: anterior/posterior (−1.3 mm), medial/lateral (+1.2 mm), and ventral to skull surface (−5.4 mm) via a 28-gauge stainless-steel cannula that stayed in the brain for 5 min after the injection before being withdrawn. Desipramine (25 mg/kg, Sigma-Aldrich), a norepinephrine reuptake inhibitor, was injected intraperitoneally 30 min prior the infusion of 6-OHDA to protect norepinephrine neurons. Following surgery, mice received 2 weeks of intensive postoperative care consisting of twice daily, 1 ml injections (i.p.) of 5% dextrose in 0.9% saline and highly palatable, high fat content food (Bacon softies, Bio-Serv) as supplementation to the normal mouse chow diet. Mice were analyzed for FFN102 imaging two weeks after the unilateral lesion.

## Imaging FFN102 transients

Sagittal slices containing the GPe were prepared as previously described with minor modifications (*Pereira et al., 2016*). Briefly, mice were killed by cervical dislocation and decapitated. Both male and female wild-type mice were used. A Leica VT1200 vibratome (Leica Microsystems, Wetzlar, Germany) was used to cut three 250 μm thick slices from each hemisphere. Slices were maintained at room temperature in oxygenated (95% O2, 5% CO2) artificial cerebrospinal fluid (ACSF [in mM]: 125.2 NaCl, 2.5 KCl, 26 NaHCO3, 0.3 KH2PO4, 2.4 CaCl2, 1.3 MgSO4, 10 glucose, 0.8 HEPES, pH 7.3–7.4, 295–305 mOsm), and used within 1 to 5 hr. Prior to imaging, each slice was incubated in an ACSF solution containing 10 μM FFN102 for 30 min. Slices were then transferred to a QE-1 imaging chamber (Warner Instruments) and held in place with a custom-made platinum wire and nylon holder. Slices were perfused with ACSF at a rate of 2–3 ml/min at room temperature (23°C).

All images were acquired using a Prairie Ultima Multiphoton Microscopy Systems (Bruker/Prairie Technologies) equipped either with a Spectra-Physics MaiTai HP DeepSee titanium-sapphire pulsed laser (Newport) and either a 10 × 1.0 NA air objective or 60 × 0.9 NA water immersion objective (Carl Zeiss Microscopy). For electrical stimulation, a twisted bipolar platinum stimulating electrode (Plastics One) was placed directly on top of the region to be imaged and pulses generated by an Iso-Flex stimulus isolator triggered through a Master-8 (each pulse 600 μs × 200 μA). Imaging was synchronized to the pulse generation using the TriggerSync software provided with the Prairie imaging system. FFN102 was excited at 760 nm and detected at 440–490 nm. For all imaging sessions, the stimulating electrode was placed using the 10x objective and then a region, approximately 50–100 μm from electrode tip, was examined under 60x at 10x digital zoom. Regions were only stimulated if they were 30 μm beneath the surface of the slice. Images, 64 × 64 pixels at 10x digital zoom (approximately 50 × 50 μm), were recorded at 10 Hz (6 μs dwell time) using a spiral scan. The use of a stimulus train provides a longer duration optical signal, and so a more robust detection of FFN release events by distinguishing the signal from rapid fluctuations. To minimize movement and deformation of slices, experiments were performed at room temperature.

## Electrical stimulation protocols for brain slices

Electrical stimulation was synchronized with frame acquisition using TriggerSync software including with the PrairieView proprietary imaging package. Electrical stimulation protocols were chosen based on the type of experiment and are detailed and justified for each experiment.

## Quantification of FFN102 fluorescence intensity

To analyze optical data, images were loaded into MATLAB and ΔF/F were extracted by first calculating the mean intensity of every image frame. To calculate baseline fluorescence, a linear fit was calculated to 500 ms of the data immediately prior to the stimulation. For each image sequence, this calculated baseline was subtracted from the fluorescence intensity over time and the resulting value divided by the baseline. This approach removes baseline activity due to bleaching or the washing out of FFN molecules. Fluctuations in baseline activity are much slower than the rise of an FFN transient, which are present at the first acquisition following the electrical stimulus. The smoothed baseline was used as the F for calculating ΔF/F for every time point of a given intensity. To calculate area under the curve values (AUC), we used the trapezoidal integration formula as implemented by MATLAB and measured the integral over frames within the stimulus period.

## Measurement of FFN in regions of interest

Previous approaches to identify FFN labeled puncta relied on localizing the centers and boundaries of puncta (*Pereira et al., 2016*). In the case of flashing FFNs, which become brighter in the extracellular space, the unit of analysis is a field of view rather than individual puncta. We classify each pixel as belonging to the edge of an FFN-filled region or background by applying a Canny-edge filter using MATLAB's *canny* function with a low threshold of 0.08 and a high threshold of 0.2. These values successfully outline the edge fluorescence from striatal images in which puncta are visible to the eye. The low threshold serves to extract even weak edges that may be present for thin varicosities or puncta with small quantities of FFN.

## Measures of calcium and DAT dependence

For $Ca^{2+}$ manipulation, slices were stimulated by using a single train of electrical pulses (10 pulses per train, 10 Hz) and imaged. The concentration of $Ca^{2+}$ was varied by switching the perfusate between 0.5, 2, and 4 mM $CaCl_2$, with a randomized order, with ten minutes between recording sessions to allow the calcium to perfuse into the slice. For the striatum, one field of view was imaged at the three concentrations and care was taken to ensure the field of view did not move during the ten minutes as the perfusate was switched. For the GPe, the same three fields of view were visited at each concentration and stimulated. The image intensity over time was obtained for each of the three stimulated fields of view, at each concentration, and an average calculated.

To measure DAT dependence in the striatum, one field of view was imaged in each slice and then nomifensine (10 μM, Sigma-Aldrich) dissolved in ACSF was perfused for ten minutes. The same field of view was then imaged with nomifensine-containing ACSF. For the GPe, ten fields of view were chosen at random, imaged, and then switched to nomifensine-containing ACSF. The same ten fields of view were visited and imaged with nomifensine-containing ACSF.

## Histology

For 6-OHDA experiments, mouse brain slices were first used in FFN102 experiments. Slices were then removed from the ACSF perfusate and placed into ice-cold 4% PFA in 0.1M TBS and stored overnight at 4°C. Slices were washed for an hour, six times, in PBS. Slices were then blocked for 2 hr with 10% fetal bovine serum, 0.5% bovine serum albumin in 0.5% TBS-Triton X-100. Primary antibody against tyrosine hydroxylase (AB152, Millipore) was applied at 1:750 dilution in block for 72 hr at 4°C. Slices were washed again for an hour, six times. Secondary anti-rabbit antibody was applied in block for 16 hr at 4°C.

For the Pitx3 (aphakia) mouse SN and VTA histology, mice under deep anesthesia were transcardially perfused with ice-cold 4% PFA in 0.1M TBS. Brains were post-fixed overnight and washed in TBS for fifteen minutes, four times. Sections of 30 μm thickness were obtained using a Leica VT2000 vibratome. The sections were incubated in blocking solution for 1 hr at room temperature, then placed in blocking solution containing primary antibody to TH at 1:750, and incubated overnight at

4°C. The sections were washed again and incubated with anti-rabbit secondary antibody in blocking solution for 1 hr at room temperature. Images for the 6-OHDA and aphakia experiments were acquired at 2.5x using a Hamamatsu camera attached to a Carl Zeiss epifluorescence microscope. MATLAB was used to process the images for level adjustment.

## Experimental design and statistical analyses

For experiments showing that FFN102 is necessary for evoking fluorescence transients (*Figure 2*), AUCs of fluorescence were measured for each slice, then slices from the two conditions were compared for significance using a t-test, and confidence intervals are also provided for comparison. To determine calcium dependence (*Figure 3*), within-slice comparisons of FFN transients were made at three separate concentrations. ANOVA was used to show dependence on calcium with each group comprising the average AUCs from a slice, obtained at different concentrations. Confidence intervals for the release at a given concentration were calculated based on the data for each group. To examine the effect of 6-OHDA treatment on the size of FFN AUC transients (*Figure 4*), we recorded from lesioned and unlesioned sides of each treated animal and compared pairs of hemispheres for each animal using a pairwise t-test. The pairwise differences were also used to compute confidence intervals to estimate the effect of depletion on the FFN signal. For aphakia animals, pairwise comparisons were made within a slice under two conditions: stimulated and not stimulated. A pairwise t-test was used to compare the two conditions and confidence intervals indicate the magnitude of the differences obtained between the two conditions for the aphakia mice. The same experiment was performed using wild-type mice, and the confidence intervals were similarly calculated. The difference between stimulated and unstimulated represented the independent samples to be compared for the aphakia and wild-type mice. Confidence intervals were used to compare these differences. To analyze the features of the striatal and GPe FFN transients (*Figures 5–7*), confidence intervals were calculated using data from multiple fields of view across multiple slices and animals. We calculated confidence intervals for comparison between striatum and GPe. For all parametric statistical tests, data were deemed normally distributed.

## Acknowledgments

We thank M Sonders for expert technical assistance. We also thank Y Schmitz for crucial advice about dopamine release measurements in the striatum, M Dunn for important discussions about FFNs, Y Schmitz and M Dunn for comments on an earlier version of this paper, and Ethan Bromberg-Martin for useful discussions about data analysis.

## Additional information

### Funding

| Funder | Grant reference number | Author |
|---|---|---|
| National Institutes of Health | T32 NS06492B-04 | Jozsef Meszaros |
| Parkinson's Disease Foundation | | Timothy Cheung<br>Un Jung Kang<br>David Sulzer |
| National Institute of Neurological Disorders and Stroke | R01 NS101982 | Un Jung Kang<br>David Sulzer |
| U.S. Department of Defense | PR161817 | Un Jung Kang |
| National Institute of Neurological Disorders and Stroke | R03 NS096494 | Un Jung Kang |
| National Institute of Mental Health | R01 MH108186 | Dalibor Sames<br>David Sulzer |
| National Institute of Mental Health | RO1 MH093672 | Christoph Kellendonk |
| JPB Foundation | | David Sulzer |

| National Institute on Drug Abuse | R01 DA07418 | David Sulzer |

The funders had no role in study design, data collection and interpretation, or the decision to submit the work for publication.

## Author contributions
Jozsef Meszaros, Conceptualization, Data curation, Formal analysis, Validation, Investigation, Visualization, Methodology, Writing—original draft, Writing—review and editing; Timothy Cheung, Conceptualization, Investigation, Writing—review and editing; Maya M Erler, Conceptualization, Data curation, Formal analysis, Investigation, Visualization, Writing—review and editing; Un Jung Kang, Dalibor Sames, Resources; Christoph Kellendonk, Conceptualization, Resources, Supervision; David Sulzer, Conceptualization, Resources, Supervision, Funding acquisition, Methodology, Writing—original draft, Writing—review and editing

## Author ORCIDs
Jozsef Meszaros (iD) http://orcid.org/0000-0002-2485-0144
Timothy Cheung (iD) http://orcid.org/0000-0002-7516-8321
Un Jung Kang (iD) http://orcid.org/0000-0002-5970-6839
Christoph Kellendonk (iD) http://orcid.org/0000-0003-3302-2188
David Sulzer (iD) http://orcid.org/0000-0001-7632-0439

## Ethics
Animal experimentation: All animal protocols followed NIH guidelines and were approved by Columbia University's Institutional Animal Care and Use Committee.

## Decision letter and Author response
Decision letter https://doi.org/10.7554/eLife.42383.019
Author response https://doi.org/10.7554/eLife.42383.020

# Additional files

## Supplementary files
• Transparent reporting form
DOI: https://doi.org/10.7554/eLife.42383.017

## Data availability
All data generated or analysed during this study are included in the manuscript and supporting files.

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
