## [Decision Letter]

[Editors’ note: a previous version of this study was rejected after peer review, but the authors submitted for reconsideration. The first decision letter after peer review is shown below.]

Thank you for submitting your work entitled "Using Fluorescent False Neurotransmitters to Characterize Exocytosis from Dopamine Synaptic Vesicles within the GPe" for consideration by *eLife*. Your article has been reviewed by two peer reviewers, and the evaluation has been overseen by a Reviewing Editor and a Senior Editor. The reviewers have opted to remain anonymous.

Our decision has been reached after consultation between the reviewers and the Reviewing Editor. Based on these discussions and the individual reviews below, we regret to inform you that your work will not be considered further for publication in *eLife*.

Although the reviewers appreciate the development of the tools to measure dopamine release in a sparsely innervated GPe area, the method, in its current form, is not sufficiently mature to characterize the properties of dopamine release and its modulation. Significant further work is needed to improve the signal-to-noise ratio of the indicator and to solidify the conclusions on the differential regulation of dopamine release in GPe versus striatum. See below to see the reviewers' comments in detail.

Reviewer #1:

In the manuscript by Meszaros et al., the authors describe their use of FFN102, a fluorescent analogue for dopamine, to characterize synaptic activity of dopaminergic neurons in the external globus pallidus. The challenge in studying dopaminergic presynaptic activity in this brain area is the scarcity of relevant synaptic terminals, which limits the applicability of other research techniques, such as cyclic voltammetry.

The authors' approach is to load brain slices with FFN102, which is loaded specifically into vesicles in dopaminergic presynaptic terminals. After washing away the extracellular FFN102, they electrically stimulated the slice and examined, using two-photon microscopy, the increase in fluorescence within two adjacent brain areas – the aforementioned GPe and the striatum, which is much more densely innervated by dopaminergic axons/terminals. The authors find that bursts of stimuli resulted in a transient increase in fluorescence in the GPe, and a much more long-lasting one in the striatum. To show that the increase is indeed the result of synaptic release of FFN102, the authors did the following controls: (1) They stimulated slices which had not been loaded with FFN102. (2) They stimulated FFN102-loaded slices in the presence of cadmium. (3) They stimulated slices incubated in various concentrations of extracellular calcium. The authors also: (4) lesioned dopaminergic neurons using 6-OHDA in one hemisphere, and (5) used a genetic model in which dopaminergic neurons fail to develop in the SNc. FFN102 fluorescence was examined in the lesioned area or in the SNc of aphakia mice.

Upon concluding that FFN102 can reveal synaptic dopaminergic activity, the authors examined differences in synaptic properties in the GPe and striatum, by (1) examining in more detail the fluorescence dynamics during the delivery of stimulation trains and (2) by examining the effect of the frequency of stimulation on the quantity of release. Finally, they examined modulation of release by D2, opiod and nicotinic receptors in the GPe. They concluded that release in the striatum is highly depressed (i.e. that it most of the difference in fluorescence is obtained during the first few stimuli), but that in the GPe it is less so (it depressed more slowly). Furthermore, GPe responses were enhanced by high-frequency stimulation. Finally, they showed that GPe responses responded to inhibition of cholinergic transmission, suggesting cholinergic modulation of presynaptic release.

1) My most significant concern relates to the assumption/claim of the authors concerning the nature of the fluorescence transient that they image. The authors indicate that fluorescence levels are integrated across the whole image, because the nature of release of FFN102 is such that fluorescence increases are not well-localized, probably because FFN102 diffuses after release. However, the authors also suggest: "FFN102 has slow reuptake kinetics and reacidification must also occur for the fluorescence to be quenched to background levels, a process that may require minutes in a heavily innervated region like the striatum." My issue with this assessment is that if the assumption is that FFN102 is released from the terminals and then diffuses into the tissue (as suggested by the need to integrate fluorescence) then I would not expect that the eventual decrease in fluorescence in the GPe would be strongly impacted by reuptake – as is implied in this paragraph. Because FFN102 is recognized specifically by DAT and VMAT, I would rather expect GPe fluorescence not to decline in this manner – because presynaptic terminals expressing DAT in this area are quite sparse (unless there is significant DAT in other locations). To address this concern, the authors could examine the effect of the acute inhibition of DAT on the GPe transients. I stress that I do not refer to experiments similar to those already reported by the authors in previous publications (Rodriguez et al., 2013), in which they showed that DAT inhibition stops the initial loading of FFN102. Rather, I would suggest inhibiting DAT in already-loaded slices and to examine the effect on the fluorescence transients after stimulation. If the difference in the fluorescence dynamics between GPe and striatum is indeed related to reuptake, I would expect DAT inhibitors to have a large effect, mostly in GPe. In the same context – the decay time of the fluorescence of FFN102 in the various figures shown in the manuscript appears to differ (for example compare Figure 3C to Figure 2A). Is the rate of fluorescence decrease relevant to the point I raised here?

2) In continuation to this point, I would expect the baseline fluorescence level in GPe to decrease successively with each given train of stimulation, if FFN102 that is loaded into vesicles is secreted and lost. In Figure 1E it is clear that some experiments behaved in this way, while others did not. Could the authors attempt to examine their data while considering this possible interpretation? What is the proportion of sessions in which fluorescence is decrease by stimulation? Is the decrease indeed related directly to stimulation?

3) I wonder about the authors' claim that Ai9 ChR2 mice have been used, while no data is presented. The ChR2 mice have the potential to increase the signal-to-noise significantly.

4) What is the sensitivity of the permutation method to noise in the imaging results? I would think that if baseline recordings are noisy, then the permutation method may increase the fraction of tests that produce a "significant" answer. Is there a difference in the noise level in the various groups of experiments?

5) The text refers to panels and data in Figure 6 (subsection “FFN102 release reveals differences in dopamine release between striatum and GPe”, first paragraph) that do not exist in the figure itself. This should be fixed before I can form an opinion on this data and its interpretation.

Reviewer #2:

In this paper, Meszaros and colleagues establish that a pH-sensitive false fluorescent neurotransmitter, FFN102, can be used to measure dopamine release in a brain area with sparse dopamine innervation. This has been a limitation in the field, and this manuscript is undoubtedly an advance. However, because the signals are small (roughly 0.3% DF/F for the signal amplitude, Figure 2A) temporal and spatial resolution remain limited. Release from single puncta cannot be measured, and action potential trains are needed to measure a signal above background. This results in a need for analyses of AUC (as opposed to amplitude) and permutation (for modulation). Although these points limit enthusiasm somewhat, the manuscript does provide a first (and likely an important) step towards better understanding of roles for dopamine in brain areas with sparse dopamine innervation.

1) FFN102 loading results in a lot of signal outside of dopamine neurons (Figure 2E). This matches well with Figure 7, where little dopamine release is detected in areas with diffuse "loading", and this diffuse lower intensity signal is also present in the striatum (Figure 2E here and Figure 2 in PMID23277566). The most likely explanation is that there is background accumulation of the dye in non-dopamine neurons. This has to be discussed and better acknowledged in the text. Furthermore, Figure 2E overlap panels are very saturated in the blue=FFN channel, different from the non-merged images that show FFN only. More careful image editing, perhaps combined with a different color choice, should be applied.

2) The experiments that address release characteristics and modulation are unclear at this point: a) The modulation experiments suggest that there is not the typical modulation of opioid and D2 receptors, and ACh receptor blockage perhaps mildly decreases the FFN102 signal during a stimulus train. However, the controls in the striatum suggest that FFN102 signals behave somewhat different from previous studies. The simple prediction from published studies would be that the first amplitude is smaller in MEC, but that dopamine release is rapidly depleted during 50 Hz/50 stimulus trains whether or not AChs are blocked. The method does not have time and spatial resolution to test this. Hence, it is possible that the differences observed have more to do with dye diffusion, clearance, etc., which could be different between the brain areas, than with release modulation.

b) Results describe experiments to suggest that "release differs between the GPe and the striatum" (section caption). Generally, better terminology to express the distinction between release and the FFN102 signal is needed. The signal is a function of release, diffusion, reuptake and reacidification. It is possible that differences in the signal arise because diffusion, reuptake etc. are different between the two brain areas, and release per se, from a bouton, is not different except that much less FFN102 is released because of the sparsity of dopamine terminals in GPe.

The meaning of these experiments is currently less clear than what the model shown in Figure 10 expresses. Leaving the data in the manuscript is fine to hint at potentially different modulation, but it may be better to remove the model and focus on the point that signals in an area with sparse dopamine innervation can be detected, rather than making strong mechanistic claims about the release.

3) The calcium dependence is shown for N = 1 slice in Figure 3C, D, this is simply below acceptable standards.

4) In Figure 7, it would be more meaningful to show a correlation between the morphological appearance (panel A) and the FFN transient (panel B). As shown, Figure 7 does not establish that areas with sparse but highly fluorescent axons release more. Furthermore, the "high" panels in A poorly reflect that point compared to the "low" panels. A better analysis is necessary.

[Editors’ note: what now follows is the decision letter after the authors submitted for further consideration.]

Thank you for resubmitting your work entitled "Evoked transients of pH-sensitive fluorescent false neurotransmitter reveal dopamine hot spots in the globus pallidus" for further consideration at *eLife*. Your revised article has been evaluated by Eve Marder (Senior Editor), a Reviewing Editor, and three reviewers.

While the manuscript has been substantially improved, some remaining important technical issues, summarized below, need to be addressed before the final decision on the manuscript is made.

Major comments:

1) All the illustrations used multiple stimuli to evoke release and the rise in fluorescence is slow and sustained. Single stimuli show rapid rise and fall of dopamine using both voltammetry and the new sensors. Thus the present results appear to be quite different. The authors should isolate responses to single stimuli (for instance by initially using sparse stimulation), followed by strong stimulation to identify the events and loci.

2) One conclusion was that the fluorescent increase in the globus pallidus was smaller and declined more rapidly. This is not surprising given that the experiments were done at room temperature. If the reuptake process plays any role in the decline in fluorescence would be inactive so the decline would be strictly dependent on diffusion. It could be that the lower concentration of FFN102 measured in the GP dropped below the detection limit thus appearing to be more rapid. Even in the hot spots of the GP the diffusion and dilution away from the point is most likely faster than might be expected from what is observed from multiple release sites in the striatum.

3) The difference in the results obtained with the paired stimulation in the GP and dorsal striatum could be accounted for by the presence of the cholinergic interneurons in the striatum that increase the probability of dopamine release dramatically such that there is substantial paired pulse depression of dopamine release.

4) There was no evidence for recovery (decrease) in fluorescence of FFN102 after cessation of stimulation in the striatum (see Figure 1). Therefore, the discussion of the authors concerning the kinetics of recovery doesn't appear to be consistent with the data (Discussion, fourth paragraph).

5) The idea of "hot spots" of release: The data presented in Figure 7 are not convincing. The authors explain that data were measured as an average of fields of view. How does this relate to the idea of hotspots? Especially when the authors refer to structures which should be of smaller dimensions than the fields. For example, in the Introduction, the authors write: "The identification of hotspots in the GPe also supports recent work showing that many striatal dopamine varicosities with clusters of synaptic vesicles are silent (Pereira et al., 2016), and this could be due to the absence or presence of local presynaptic scaffolding proteins (Liu et al., 2018)." However, the dimensions of the so-called "hot spots", as measured in the current study, are significantly larger than what one would expect varicosities to be. Therefore, such a suggestion as to the biological basis for the hotspots does not appear plausible. Later, the authors write: "For example FFN transients could be used to locate dopamine release near specific cell types, such as the arkypallidal cells which project back into the striatum and the protopallidal cells which project downstream to the SNr and GPi (Gittis et al., 2014; Mastro et al., 2014; Hernández et al., 2015)." Are the dimensions of the hotspots small enough to find specific cells within slices? If the authors measure transients from whole fields, then this does not appear to be the case.

---

## [Author Response]

[Editors’ note: the author responses to the first round of peer review follow.]

We have now conducted a year’s worth of additional experiments and analysis that we believe fully answers all of the critiques, and we request that you reconsider if this can be reviewed again by the reviewers. Both Reviewers however expressed insightful critiques/concerns, and so we have carried out substantial new experiments and revised the study as they recommended.

[…] 1) My most significant concern relates to the assumption/claim of the authors concerning the nature of the fluorescence transient that they image. The authors indicate that fluorescence levels are integrated across the whole image, because the nature of release of FFN102 is such that fluorescence increases are not well-localized, probably because FFN102 diffuses after release. However, the authors also suggest: "FFN102 has slow reuptake kinetics and reacidification must also occur for the fluorescence to be quenched to background levels, a process that may require minutes in a heavily innervated region like the striatum." My issue with this assessment is that if the assumption is that FFN102 is released from the terminals and then diffuses into the tissue (as suggested by the need to integrate fluorescence) then I would not expect that the eventual decrease in fluorescence in the GPe would be strongly impacted by reuptake – as is implied in this paragraph. Because FFN102 is recognized specifically by DAT and VMAT, I would rather expect GPe fluorescence not to decline in this manner – because presynaptic terminals expressing DAT in this area are quite sparse (unless there is significant DAT in other locations). To address this concern, the authors could examine the effect of the acute inhibition of DAT on the GPe transients. I stress that I do not refer to experiments similar to those already reported by the authors in previous publications (Rodriguez et al., 2013), in which they showed that DAT inhibition stops the initial loading of FFN102. Rather, I would suggest inhibiting DAT in already-loaded slices and to examine the effect on the fluorescence transients after stimulation. If the difference in the fluorescence dynamics between GPe and striatum is indeed related to reuptake, I would expect DAT inhibitors to have a large effect, mostly in GPe. In the same context – the decay time of the fluorescence of FFN102 in the various figures shown in the manuscript appears to differ (for example compare Figure 3C to Figure 2A). Is the rate of fluorescence decrease relevant to the point I raised here?

We thank the reviewer for this insightful analysis, and agree that as a DAT substrate, it could be that reuptake might acutely inhibit the decay of the signal or increase its amplitude, and the decay times are indeed relevant to these points.

We therefore conducted additional experiments in which we acutely apply nomifensine into the FFN102 loaded slices We found that DAT inhibition neither increased the amplitude nor decay of the GPe transients. Thus, the FFN “flash” is a measure of release but not of acute reuptake. The new data are displayed in Figure 6.

2) In continuation to this point, I would expect the baseline fluorescence level in GPe to decrease successively with each given train of stimulation, if FFN102 that is loaded into vesicles is secreted and lost. In Figure 1E it is clear that some experiments behaved in this way, while others did not. Could the authors attempt to examine their data while considering this possible interpretation? What is the proportion of sessions in which fluorescence is decrease by stimulation? Is the decrease indeed related directly to stimulation?

We certainly agree that as the synaptic vesicles fuse, that there is less FFN to release with subsequent stimuli, and analysis of FFN destaining in previous papers has used this form of analysis. It is less relevant for FFN102 in the present approach, as we are taking advantage of the design feature that it becomes brighter upon release due to the shift in pH. This signal is measureable even following a short electrical stimulation, one second as compared with minutes for previous studies. The major consequence of release is therefore the transient which is a much larger signal, particularly in this very sparsely innervated region, in which the low signal of FFN within the acidic synaptic vesicles in thin axons is undetectable above background levels. Indeed, we now also show that in the GPe, there is no correlation between puncta and amount of release in Figure 7B.

To make the point more clearly that this novel approach takes advantage of a transient flash of release rather than the previous FFN approaches that analyzed destaining from axonal varicosities, we changed the title of the manuscript and state in the Abstract:

“we introduce an optical approach using a pH-sensitive fluorescent false neurotransmitter, FFN102, that exhibits increased fluorescence upon exocytosis from the acidic synaptic vesicle to the neutral extracellular milieu.”

In the Introduction:

“In this study, we adapt a pH-sensitive FFN, FFN102, which is a substrate for the dopamine transporter, DAT, and the vesicular monoamine transporter, VMAT2 (Rodriguez et al., 2013) as the first use of a “flashing FFN”. The signal from FFN102 is well suited for study synaptic release in sparsely innervated regions as it is brighter in the neutral extracellular milieu than the acidic milieu of the synaptic vesicle lumen (Lee et al., 2010).”

In the Results:

“We then addressed whether the differences in kinetics and size of FFN transients between the striatum and the GPe might result from differential reuptake, which might broaden the signal. […] Neither the decay time to its half-maximum value (Figure 6E) nor the shape of the decay were altered by nomifensine (Figure 6F), indicating a lack of a role for DAT in altering evoked FFN102 signals.”

And to discuss that the amount remaining in the axons is a less reliable indicator for very sparsely innervated regions we write:

“The greater release of FFN102 in GPe at higher frequencies (Figure 5) suggested that GPe dopamine axons might have the capacity to generate large FFN transients. […] Surprisingly, GPe regions with large FFN102 transients did not have obvious FFN102 puncta within the field of view (Figure 7B).”

In the Discussion:

“Here we introduce a new use for pH sensitive FFNs, which is to resolve release from dopamine axons in a region of very sparse innervation by “FFN flashes”, i.e., short duration calcium-dependent release transients during exocytosis as the fluorescence is unquenched upon exocytosis from the acidic vesicle lumen to the neutral extracellular space. FFN102, a DAT and VMAT2 substrate with a pK of 6.2 (Lee et al., 2010), is well suited for this application, as its signal increases nearly 400% between the acidic synaptic vesicle (~pH 5.6) and the extracellular milieu at pH 7.4 (Rodriguez et al., 2013).”

3) I wonder about the authors' claim that Ai9 ChR2 mice have been used, while no data is presented. The ChR2 mice have the potential to increase the signal-to-noise significantly.

We have deleted mention of the Ai9-ChR2 mice, as while the localization can be used, in our current approach, the LED light contaminated the FFN signal. We think that ongoing development of red-shifted photoactivatable channels by other research groups will be used for this purpose, but they are not yet available to us, and so we have no useful data on this approach.

4) What is the sensitivity of the permutation method to noise in the imaging results? I would think that if baseline recordings are noisy, then the permutation method may increase the fraction of tests that produce a "significant" answer. Is there a difference in the noise level in the various groups of experiments?

There is little difference in the noise level in the experiments, although there is far more baseline in the striatum than the GPe. We now analyze the data without the permutation method as written in the Materials and methods:

“To calculate baseline fluorescence, a linear fit was calculated to 500 ms of the data immediately prior to the stimulation. […] The smoothed baseline was used as the F for calculating ΔF/F for every time point of a given intensity.”

Other concerns this reviewer had with figures relating to hotspot analysis and modulation have been removed. A more detailed methodology has been used to assess hotspots. The possibility of measuring modulation is taken up in the Discussion.

Additionally, we have made a correction to the placement of the discussion related to Figure 6 (Critique 5).

5) The text refers to panels and data in Figure 6 (subsection “FFN102 release reveals differences in dopamine release between striatum and GPe”, first paragraph) that do not exist in the figure itself. This should be fixed before I can form an opinion on this data and its interpretation.

See response to comment #4.

Reviewer #2:[…] 1) FFN102 loading results in a lot of signal outside of dopamine neurons (Figure 2E). This matches well with Figure 7, where little dopamine release is detected in areas with diffuse "loading", and this diffuse lower intensity signal is also present in the striatum (Figure 2E here and Figure 2 in PMID23277566). The most likely explanation is that there is background accumulation of the dye in non-dopamine neurons. This has to be discussed and better acknowledged in the text. Furthermore, Figure 2E overlap panels are very saturated in the blue=FFN channel, different from the non-merged images that show FFN only. More careful image editing, perhaps combined with a different color choice, should be applied.

This critique is related to Critique 2 by reviewer 1, please also see that discussion.

We have analyzed the relationship between “puncta”, i.e., static fluorescent areas and FFN release and find that in the very sparsely innervated GPe, there is no correlation between puncta and amount of release (Figure 7B).We now write: “The greater release of FFN102 in GPe at higher frequencies (Figure 5) suggested that GPe dopamine axons might have the capacity to generate large FFN transients. […] Surprisingly, GPe regions with large FFN102 transients did not have obvious FFN102 puncta within the field of view (Figure 7B).”

This may be related to the lack of significant synaptic vesicle clusters in this region of the axon, in very strong contrast to the same axons in the striatum, which have many distinctive presynaptic sites. We now write in the Discussion:

“On average, most GPe regions showed release after the first pulse, but the amount of evoked release was strikingly variable, and far more responsive to increased stimulus frequency than release in the striatum. […] The lower evoked FFN transients in GPe may be due to exocytosis from comparatively small reservoirs of synaptic vesicles in thin portions of the axon that do not maintain large vesicle clusters.”

We agree with the reviewer that diffuse loading is a concern, but note that 1) DAT blockade 2) the aphakia mutation 3) 6OHDA, nearly abolished FFN102 signal in both striatum and GPe. Thus, the non-dopaminergic component of signal in both brain regions is minor. We now write in the Results section:

“To examine if FFN102 is loaded in GPe axons as a substrate for the dopamine transporter (DAT), we co-incubated slices with both FFN102 and the DAT inhibitor, nomifensine, for 30 minutes before the stimuli. The significant decrease of evoked fluorescent transients in slices co-incubated with nomifensine is consistent with an inhibition of uptake and loading of FFN102 into DA axons (Figure 2C, D).”

“To confirm whether dopaminergic neurons are the source of the FFN transients, we used the toxin 6-hydroxydopamine (6-OHDA lesion) to unilaterally lesion the dopamine projections passing through the medial forebrain bundle. […] Thus, SNc dopamine neurons are the primary contributor to FFN102 release in the GPe.”

2) The experiments that address release characteristics and modulation are unclear at this point: a) The modulation experiments suggest that there is not the typical modulation of opioid and D2 receptors, and ACh receptor blockage perhaps mildly decreases the FFN102 signal during a stimulus train. However, the controls in the striatum suggest that FFN102 signals behave somewhat different from previous studies. The simple prediction from published studies would be that the first amplitude is smaller in MEC, but that dopamine release is rapidly depleted during 50 Hz/50 stimulus trains whether or not AChs are blocked. The method does not have time and spatial resolution to test this. Hence, it is possible that the differences observed have more to do with dye diffusion, clearance, etc., which could be different between the brain areas, than with release modulation.b) Results describe experiments to suggest that "release differs between the GPe and the striatum" (section caption). Generally, better terminology to express the distinction between release and the FFN102 signal is needed. The signal is a function of release, diffusion, reuptake and reacidification. It is possible that differences in the signal arise because diffusion, reuptake etc. are different between the two brain areas, and release per se, from a bouton, is not different except that much less FFN102 is released because of the sparsity of dopamine terminals in GPe.The meaning of these experiments is currently less clear than what the model shown in Figure 10 expresses. Leaving the data in the manuscript is fine to hint at potentially different modulation, but it may be better to remove the model and focus on the point that signals in an area with sparse dopamine innervation can be detected, rather than making strong mechanistic claims about the release.

We agree with these critiques, and have deleted the model and analysis of modulation. We further agree that as the reviewer writes, a better terminology is needed. We now use the term “flash”, i.e., the evoked FF102 transient. As discussed in the DAT blockade experiments, the flash is now more thoroughly explained as being due to release and the consequent increase in emissia due to the pH shift, and the decay is due to diffusion, whereas reacidification and reuptake do not play significant roles during the rapid transient events.

3) The calcium dependence is shown for N = 1 slice in Figure 3C, D, this is simply below acceptable standards.

We have repeated all of the calcium experiments with N=9 slices in the striatum and N=16 slices in the GPe. We performed within-slice comparisons at three concentrations to obtain data for these experiments, which are in the revised Figure 3.

4) In Figure 7, it would be more meaningful to show a correlation between the morphological appearance (panel A) and the FFN transient (panel B). As shown, Figure 7 does not establish that areas with sparse but highly fluorescent axons release more. Furthermore, the "high" panels in A poorly reflect that point compared to the "low" panels. A better analysis is necessary.

We have followed the reviewer’s advice and added significant new analysis, and indeed the more sparsely innervated regions with highly fluorescent axons do not release more. As discussed above, the low level of signal in the synaptic vesicle prior to release means that the flash is more important for this analysis than the background levels. This is we think a facet of measuring in very sparsely innervated regions. The new data are in Figure 7. We now write:

“The greater release of FFN102 in GPe at higher frequencies (Figure 5) suggested that GPe dopamine axons might have the capacity to generate large FFN transients. […] We then determined each field of view’s derivative value before stimuli, after a single pulse, and after two pulses (Figure 7H). For both the striatum and the GPe, the largest transient occurred at the first electrical pulse.”

[Editors' note: the author responses to the re-review follow.]

Major comments:1) All the illustrations used multiple stimuli to evoke release and the rise in fluorescence is slow and sustained. Single stimuli show rapid rise and fall of dopamine using both voltammetry and the new sensors. Thus the present results appear to be quite different. The authors should isolate responses to single stimuli (for instance by initially using sparse stimulation), followed by strong stimulation to identify the events and loci.

We believe that the impression of apparently slow kinetics of the dopamine signal and differences between single and train stimuli is a misunderstanding and try to correct it here: the FFN102 signal rise/release is not slower than results using cyclic voltammetry in the striatum, and in the GPe is not sustained.

As shown in Figure 1 of (Schmitz et al., 2001)], with voltammetry, the rise in the striatum occurs at about 180 msec (dotted line in the figure), and the sampling using this technique is typically 10 Hz, which is at or near the effective limit due to the charging “Faradaic” current which produces excessive background at higher charging rates. In this older study from our group, we showed that the cyclic voltammetry mode of electrochemistry is more accurate for determining release and reuptake than amperometry, as the second approach consumes dopamine at the electrode surface that creates a local concentration minimum and removes local extracellular dopamine faster than the DAT, whereas cyclic voltammetry regenerates the dopamine, and neither consumes it nor interferes with the kinetic analysis of DAT. The consuming face also artificially decreases the rise time of the signal by vastly increasing the local concentration gradient. Therefore, the estimate of ~180 msec for the rising phase of dopamine by a single pulse in the striatum is to our knowledge the best estimate of response to evoked release by electrical stimuli with extant techniques. Note also that the 180 msec is far longer than the release from synaptic vesicle fusion which is less than 1 msec (Staal et al., 2004), and is due to the overflow/diffusion of dopamine from multiple release sites to the carbon fiber: this is however the situation for dopamine neurotransmission in the striatum and other regions as the dopamine receptors are extrasynaptic and are activated at this time scale.

This means that with FFN102, the 10 pulse 10 Hz stimuli and measurements at 10 Hz that the rise time is essentially identical to cyclic voltammetry. This may be clearest to see in the present study’s Figure 3, which shows a rapid increase in slope by the second pulse for both the striatum (A) and GPe (C), i.e., by 200 msec, and that with the sampling we are using, it is at the limit of resolution. It is true that FFN optical signals do not have the excellent level of low noise and signal resolution of carbon fiber electrochemistry, but the GPe signal is too small for electrochemistry to measure at all. The FFN signal is in contrast sufficient to clearly indicate that the rise time is also about 200 msec, essentially identical to voltammetry.

We also analyze the kinetics of the rising phase by reporting the derivative, i.e., the slope, in Figure 7H that shows that both for striatum and GPe, nearly all of the rising phase is due to the release from the first stimuli. Figure 7G shows a higher derivative of the first pulse from the striatum than GPe, but this is because as the signal is higher in the striatum, the rising slope is as well. It may be that the impression that this is a slow rise is because we did not make it sufficiently clear that with the spiral scan used in this study, each data point requires 100 msec, that is we can only measure a change at 10 Hz intervals. This is quite rapid for optical methods in the brain slice, and very slow for electrophysiology, and the same sampling rate used for cyclic voltammetry.

We trigger the start of the image acquisition and the first pulse simultaneously, and so the response to the first pulse is complete only during the second point, when the spiral scan returns to the start point. The rise time may be faster than 100 msec following the stimulus, but the limited effective sampling of the spiral scan in the slice preparation, in comparison to electrophysiology, limits us to resolve to that duration.

Finally, the rationale for using trains of pulses rather than a single pulse, even though most of the release occurs in response to the first pulse is because of the high noise in optical measurements (in comparison to electrochemistry). With the pulse train, the signal remains elevated for over a second, which provides 10 data points, and this helps to determine that a “random” fluctuation in the light level is not misinterpreted as an FFN signal.

In the Materials and methods, we had previously written and retain the statement:

“Images.. were recorded at 10 Hz (6 µs dwell time) using a spiral scan.”

and now add:

“The use of a stimulus train provides a longer duration optical signal, and so a more robust detection of FFN release events by distinguishing the signal from rapid fluctuations.”

To make it clear that there is a rapid increase in the rise time, we now write:

“While we only resolve signals at 100 msec intervals, it is apparent that for both the striatum and the GPe, the largest contribution to the total transient occurred at the first electrical pulse and is consistent with estimates of the rising phase using cyclic voltammetry, which is ~180 msec in the striatum (Schmitz et al. 2001).”

Regarding the kinetics of the signal decay, please note that the decay in Figure 1A is much shorter for the GPe than for the striatum, and that in Figure 2C, the falling phase is complete within a second after the stimulus: if we could resolve it better, we suspect that it would be well fit by an exponential decay, consistent with diffusion from a point source. As we hope to have made clear in the present paper, dopamine in the striatum is mostly cleared by DAT, and while FFN102 is a DAT substrate it is of much lower affinity than dopamine itself and not rapidly cleared, so that diffusion plays a bigger role. We believe that the reliance on diffusion for the decay of the FFN102 signal is confirmed in Figure 6, which shows that nomifensine does not increase the duration of the signal. In the GPe, there is very little DAT activity as the dopamine axons are very sparse, and so diffusion plays a greater role: however, there is also less FFN released, and so the signal is of much shorter duration than for the striatum. We note:

“The longer fluorescence decay in striatum is in part due to diffusion from out-of-plane striatal terminals, whereas in the GPe, areas of high release are few and spatially separated, and so less signal would diffuse from distal release sites to the regions of interest (Sulzer and Pothos, 2000).”

In sum, we hope that by now clearly stating that the kinetics of the rising phase of the FFN signal is consistent with that of cyclic voltammetry that the issue is more completely analyzed.

2) One conclusion was that the fluorescent increase in the globus pallidus was smaller and declined more rapidly. This is not surprising given that the experiments were done at room temperature. If the reuptake process plays any role in the decline in fluorescence would be inactive so the decline would be strictly dependent on diffusion. It could be that the lower concentration of FFN102 measured in the GP dropped below the detection limit thus appearing to be more rapid. Even in the hot spots of the GP the diffusion and dilution away from the point is most likely faster than might be expected from what is observed from multiple release sites in the striatum.

This critique is very closely related to above discussion. We agree with all of these points: diffusion indeed appears to be responsible for the rate of the falling phase, and this is experimentally consistent with the nomifensine experiments in Figure 6, demonstrating that under these conditions that there is very little effect of DAT reuptake on FFN102. We further agree that there is indeed more FFN102 released from the multiple sites in the striatum, and that this is why diffusion appears “faster” in the GPe than striatum: more accurately, diffusion is a rate δC (concentration)/δx (distance) governed by the concentration gradient, not a velocity and so “faster” is a misnomer. We of course agree that the signal decay is indeed faster and that the higher concentration in the striatum leads to an apparently much “slower diffusion” to reach the detection limit.

We also agree with the implication that it is preferable in principle to perform experiments at physiological temperature, but the motion artifacts preclude this with our current techniques: we hope that we and the field will be able to overcome this limitation. To acknowledge the point about the contribution of diffusion to the decay phase of the signal in the GPe vs. striatum, in addition to the statement to Critique 1, we write:

“The decay of FFN102 transients below the apparent detection limit was much faster in the GPe than the striatum, including in GPe hotspots with high evoked transient amplitudes similar to the striatum. […] The longer fluorescence decay in striatum is in part due to diffusion from out-of-plane striatal terminals, whereas in the GPe, areas of high release are few and spatially separated, and so less signal would diffuse from distal release sites to the regions of interest (Sulzer and Pothos, 2000).”

3) The difference in the results obtained with the paired stimulation in the GP and dorsal striatum could be accounted for by the presence of the cholinergic interneurons in the striatum that increase the probability of dopamine release dramatically such that there is substantial paired pulse depression of dopamine release.

We agree in part, and the reviewer is likely referring to multiple studies by Stephanie Cragg’s group, Joseph Cheer, as well as a study by Melchoir et al. from Sara Jones’s lab (2015) demonstrating that within the striatum, activation of cholinergic receptors dramatically enhances dopamine releases after a single pulse of electrical stimulation (Rice and Cragg, 2004; Cachope et al., 2012; Melchior et al., 2015). Please note that the Melchoir et al. study showed that a train of 20 electrical pulses, similar to the 10 pulse protocol we employ here, produced the same amount of dopamine release whether or not DHbE was present. This also relates to an early study by our group showing that the effect of nicotine and antagonists on evoked dopamine release are overcome by higher frequency activity (Zhang and Sulzer, 2004), and independently by Rice and Cragg (Rice and Cragg, 2004). To our knowledge, all of the extant work in the field indicates that cholinergic receptors exerts relatively little effect on the total amount of dopamine released during a train of pulses.

We now write:

“Local interactions with cholinergic interneurons may contribute to differences in the FFN signals between striatum and GPe, although these effects are minimized with the train stimuli used in this study (Melchoir et al. 2015, Zhang and Sulzer 2004, Rice and Cragg, 2004).”

4) There was no evidence for recovery (decrease) in fluorescence of FFN102 after cessation of stimulation in the striatum (see Figure 1). Therefore, the discussion of the authors concerning the kinetics of recovery doesn't appear to be consistent with the data (Discussion, fourth paragraph).

Thank you for pointing this out, we had given a mistaken impression. There is a recovery of fluorescence of FFN102 in the striatum, but it is very slow, as shown here from Figure 5 of Rodriguez et al., 2012. We now revise the former statement to read:

“The decay of FFN102 transients below the apparent detection limit was much faster in the GPe than the striatum, which requires several minutes (Rodriguez et al., 2012) including for the GPe hotspots with high evoked transient amplitudes similar to the striatum.”

5) The idea of "hot spots" of release: The data presented in Figure 7 are not convincing. The authors explain that data were measured as an average of fields of view. How does this relate to the idea of hotspots? Especially when the authors refer to structures which should be of smaller dimensions than the fields. For example, in the Introduction, the authors write: "The identification of hotspots in the GPe also supports recent work showing that many striatal dopamine varicosities with clusters of synaptic vesicles are silent (Pereira et al., 2016), and this could be due to the absence or presence of local presynaptic scaffolding proteins (Liu et al., 2018)." However, the dimensions of the so-called "hot spots", as measured in the current study, are significantly larger than what one would expect varicosities to be. Therefore, such a suggestion as to the biological basis for the hotspots does not appear plausible. Later, the authors write: "For example FFN transients could be used to locate dopamine release near specific cell types, such as the arkypallidal cells which project back into the striatum and the protopallidal cells which project downstream to the SNr and GPi (Gittis et al., 2014; Mastro et al., 2014; Hernández et al., 2015)." Are the dimensions of the hotspots small enough to find specific cells within slices? If the authors measure transients from whole fields, then this does not appear to be the case.

Thank you for this comment, and we were not sufficiently clear in the previous version. We indeed analyze the increased fluorescent signal over an entire field of view, which is 30 x 30 µm: presynaptic varicosities in the striatum labeled with FFN are close to 1-2 µm, and the field of view is close to the scale of one or two cell bodies. Therefore, we can only provide an upper limit of a “hot spot” in a region.

Nevertheless, please note that in Figure 7E, fields of view with high release (yellow circles) are neighbored by and even overlap with fields of view with dramatically less release in a representative slice. This is analyzed in Figure 7F, which shows that the spatial distribution of release sites was not concentrated in clusters of “hotspots”, and that they show no more tendency to be close together than other sites. Thus, a “hotspot” in one field of view did not make it any more likely to find one in a nearby field of view. Given that we do localize “hotspots” to regions the size of 1-2 cell bodies, we think this is small enough to be close to particular cells within the slice or near specific populations of dendrites. To make this point more clearly, we now write:

“The current approach is able to resolve FFN transients in 30 x 30 µm fields of view, close to the size of neuronal cell bodies, and so FFN transients might be used to characterize dopamine release near specific cell types, such as the arkypallidal cells which project back into the striatum and the protopallidal cells that project to the SNr and GPi (Gittis et al., 2014; Mastro et al., 2014; Hernández et al., 2015).”